# Single-cell transcriptome sequencing for opening the blood-brain barrier through specific mode electroacupuncture stimulation

Congcong Ma[1,2,3†], Zhaoxing Jia[2†], Tianxiang Jiang[2†], Qian Cai[2†], Jinding Yang[2†], Lin Gan[2], Kecheng Qian[2], Zixin Pan[2], Qinyu Ye[2], Mengyuan Dai[4], Xianming Lin[1,2,3]*

[1]The Third Affiliated Hospital of Zhejiang Chinese Medical University, Moganshan, China; [2]The Third Clinical Medical College, Zhejiang Chinese Medical University, Hangzhou, China; [3]Key Laboratory of Acupuncture and Neurology of Zhejiang Province, Hangzhou, China; [4]Department of Rehabilitation, Lishui Central Hospital, Lishui, China

## eLife Assessment

This study presents a **valuable** finding that the blood-brain barrier (BBB) may be modulated through specific modes of electroacupuncture stimulation. The data were collected and analyzed using a **solid** and validated methodology, and can be used as a starting point for functional studies of the BBB for drug delivery across healthy and diseased states. The work will be of broad interest to scientists working in the field of drug delivery and drug development.

**Abstract** The blood-brain barrier (BBB) interferes with the treatment of central nervous system disorders owing to the complexity of its structure and restrictive function. Thus, it is challenging to develop central nervous system drug delivery strategies. Specific mode electroacupuncture (EA) stimulation can effectively open the BBB in rats. Here, we used single-cell RNA sequencing (scRNA-seq) to comprehensively map the cell population in the Sprague-Dawley rat cerebral cortex. We identified 23 cell subsets and eight types of cells in the brain by cell annotation. scRNA-seq revealed transcriptional changes in the cerebral cortex under EA. Our findings offer valuable insights into the molecular and cellular modifications in the brain resulting from EA intervention and serve as a resource for drug delivery across healthy and diseased states. Innovative approaches to enhance BBB opening will lead to more effective therapeutic plans and enhanced drug delivery.

## Introduction

The blood-brain barrier (BBB), which is crucial for drug delivery to the brain, mainly comprises specialized brain endothelial cells with tight junctions, a basement membrane, neurons, pericytes, and astrocytes. Its chemical composition and physical properties create an adaptable interface. Single-cell RNA sequencing (scRNA-seq) can systematically identify cell types and post-intervention molecular changes by analyzing the gene expression profiles of individual cells. This enables researchers to infer cell-type-specific interactions and predict the molecular basis and functional significance of these ligand-receptor interactions.

*For correspondence:
linxianming1966@163.com

†These authors contributed equally to this work

Competing interest: The authors declare that no competing interests exist.

Our previous studies have shown that specific mode electroacupuncture (EA) at head stimulation points (Baihui and Shuigou acupoints), using parameters of '2/100 Hz, 3 mA, 6 s-6 s, 40 min,' can produce a significant and effective BBB opening effect, promoting the entry of Evans blue and FITC-Dextran of different molecular weights into the brain, with the permeability effect being more prominent in the frontal lobe region. (*Zhang et al., 2018*; *Zhang et al., 2020*; *Ma et al., 2022*; *Lin et al., 2023*; *Ma et al., 2024*; *Dai et al., 2025*). In addition, we found that EA-induced BBB opening exhibits stimulus dependency and a time window effect. However, the mechanisms underlying EA-induced BBB opening remain largely unknown. Therefore, based on the clear effects established in our previous research, this study investigates transcriptomic changes in the Sprague-Dawley rat frontal cortex following EA intervention, in order to preliminarily characterize possible underlying mechanisms.

We addressed this challenge by presenting a comprehensive single-cell molecular map of the rat frontal cortex, which covers approximately 98,338 cells from 10 samples, including freshly excised cortical brain tissue after EA intervention and responsive tissue from the control group. We characterized 23 clusters in the rat frontal cortex, including eight annotated cell types.

## Methods

### Animal model

Male Sprague–Dawley rats (Specific Pathogen Free), 8–10 weeks old and weighing 180–220 g, were obtained from Shanghai Sipul-Bike Experimental Animal Company. All rats were housed in the Zhejiang Chinese Medical University Laboratory Animal Research Center under a 12 hr light/dark cycle with food and water given ad libitum. Rats were group-housed and were randomly assigned to experimental groups.

### Materials and methods

#### Grouping and interventions

Rats were randomly assigned to the EA and CON groups (n=5 per group). The EA group received EA; the CON group received the same non-stimulated binding treatment. Sterile disposable needles from Beijing Zhongyan Taihe Medical Equipment Co., Ltd. (Beijing, China) were used for EA. The GV20 (Baihui) and GV26 (Shuigou) acupuncture points were selected, with needle dimensions of 25 mm × 0.13 mm for GV20 and 16 mm × 0.07 mm for GV26. A custom relay, connected to an acupuncture point stimulator (HANS-200A, Nanjing Jisheng Co., Ltd., Nanjing, China), activated the needles in a 6 s on/off cycle. The current was set at 3 mA at a frequency of 2/100 Hz, and the stimulation lasted for 40 min.

#### Cerebral cortex issue preparation

At the conclusion of the experiments, rats were anesthetized using intraperitoneally administered pentobarbital (50 mg/kg). Subsequently, normal saline solution was circulated through the left side of the heart. The brain was removed via decapitation, and the frontal cortex tissue was immediately separated and placed in an ice-cold sterile RNase-free culture dish with an appropriate amount of calcium-free and magnesium-free 1× phosphate-buffered saline (PBS). The tissue was then sliced into 0.5 mm$^3$ pieces and washed with 1×PBS. Non-target tissues, such as the blood cells and brain white matter, were carefully removed.

Single cells were isolated using a previously described method (*Chi et al., 2024*). Standard cDNA amplification and library construction were performed. The libraries were sequenced on an Illumina NovaSeq 6000 system (paired-end, 150 bp) by LC-BioTechnology Co., Ltd. (Hangzhou, China) with a minimum depth of 20,000 reads per cell.

### Quantification and statistical analysis

#### Bioinformatics analysis

Sequencing results were demultiplexed and converted to the FASTQ format using Illumina bcl2fastq (version 2.20). The Cell Ranger pipeline (version 6.1.1) handles sample demultiplexing, barcode processing, and single-cell 3' gene counting. From five EA and five CON samples, 98,338 single cells were captured using the 10X Genomics Chromium Single Cell 3' solution. The output was analyzed

using Seurat (version 3.1.1) for dimensional reduction, clustering, and scRNA-seq data analysis. Quality control filtered cells with fewer than 500 expressed genes, UMI counts below 500, and >25% mitochondrial DNA-derived gene expression; as a result, 80,713 cells were included in the analysis.

To enhance data visualization, we used Seurat for the dimensionality reduction of all cells, followed by t-distributed stochastic neighbor embedding (t-SNE) for two-dimensional projection. The process included (1) applying the LogNormalize method from Seurat's 'Normalization' function to determine the gene expression values; (2) conducting principal component analysis (PCA) of normalized expression values, selecting the top 10 principal components (PCs) for clustering and t-SNE analysis; (3) using the weighted shared nearest neighbor (SNN) graph-based clustering method to identify clusters; and (4) selecting marker genes for each cluster using the Wilcoxon rank-sum test (default parameters 'bimod' and the likelihood ratio test) via Seurat's FindAllMarkers function. Marker genes were chosen based on their expression in more than 10% of cells within a cluster, with an average log value (fold change) greater than 0.25 (default parameter: 0.26).

Bioinformatics analysis was performed using the OmicStudio tools, which are accessible at https://www.omicstudio.cn/tool.

## Doublet removal

Occasionally, two or more cells are encapsulated in the same gem during single-cell sequencing, resulting in abnormal barcodes that could affect the analysis (called doublets or multilets). Therefore, we used Doublet Finder software to identify and remove double and multiple cells to ensure the quality of the results, with the 7.5% doublet formation rate guideline recommended by 10X Genomics for single-cell analysis, aiming to identify the most likely doublets.

## Classification of single-cell clusters

The single-cell subset classification analysis involved several steps. First, low-quality cells were removed, and expression values were normalized using the Log Normalize method in Seurat's 'Normalization' function. Next, PCA was performed on the normalized expression values to reduce the dimensionality and variables. The top 10 PCs were selected for clustering and subpopulation analyses. Finally, the Seurat software used a graph-theory-based clustering algorithm to divide the cells. This process included constructing a k-nearest neighbor clustering graph based on the Euclidean distance from significant PCs, optimizing intercellular distance weights using the Jaccard similarity and identifying cell clusters using an SNN module-optimized clustering algorithm.

## Analysis of marker genes

Seurat used a bimodal likelihood ratio statistical test to analyze differentially expressed genes (DEGs) across various cell populations to identify genes that were upregulated in different clusters. The screening criteria for upregulated genes were as follows: (1) genes expressed in more than 10% of cells in the target or control subpopulations; (2) p-value≤0.01; and (3) gene expression fold change (logFC) ≥0.26, meaning that the fold increase in gene expression was ≥$2^{0.26}$.

## Cell-type annotation

Cell annotation results were defined using SingleR by employing a reference dataset derived from Mouse RNAseq Data. The correlation between the gene expression profiles of each cell and the dataset was calculated by assigning each cell a corresponding identification category. We also tallied the cell types identified within each cluster and their respective percentages, selecting the cell type with the highest correspondence for each cluster as the cell-type identification result.

## Differential gene expression analysis

For differential analysis, the bimod algorithm was chosen, and the following parameters for screening DEGs were used: (1) p<0.01, (2) logFC ≥0.26, and (3) the threshold for gene expression in at least one type of cell was 0.1.

## Functional enrichment analysis

For the D significance enrichment analysis, we initially mapped all DEGs that were differentially expressed by various terms. We then calculated the number of genes belonging to each term and

used hypergeometric tests to identify Gene Ontology entries significantly enriched in DEGs compared to the entire genome background. The Kyoto Encyclopedia of Genes and Genomes (KEGG) enrichment analysis followed the same strategy. Enrichment scatter plots and bar charts were generated using OmicStudio tools at https://www.omicstudio.cn/tool.

### Cell-cell communication analysis

CellPhoneDB was used to analyze cellular communication by predicting ligand-receptor interactions based on receptor expression in one cell type and ligand expression in another. For each gene, we calculated the percentage of cells expressing and the average expression of the genes. Ligand-receptor pairs were included only if the proportion of cells expressing both genes exceeded 10%. For the polymers, the subunit with the lowest average expression represented the receptor in the statistical analysis. We compared all cell types in pairs within the filtered dataset by randomly arranging the cells into a new population with a default of 1000 arrangements. We determined the mean expression of ligands in a randomly arranged cell population and the average expression of receptors in interacting cell types and then calculated their mean. This process was repeated several times to obtain a normal mean distribution. Next, we calculated the actual average number of ligand-receptor pairs in the original cell population. The significance of the receptor-ligand pair was determined by the proportion of the calculated average values equal to or exceeding the actual average. We then sorted the interactions between cell types based on the number of significant ligand-receptor pairs for manual screening of biologically relevant interactions. As CellPhoneDB only recognizes human genes or proteins, we used the Ensemble tool to convert genes into their homologous counterparts before conducting further analyses.

## Results

### ScRNA-seq and clustering analysis

We used 10X Genomics technology to analyze the mRNA levels of individual cells isolated from the frontal cortex of rats (*Figure 1a*). We analyzed 98,338 individual cells from 10 different biological samples. A clear boundary between cells and non-cells was visible. The environmental background was very clean, with almost no environmentally free RNA (*Figure 1b*). We eliminated cells with potential double droplets, characterized by an unusually high number of detected genes, and unhealthy cells with typical mitochondrial mRNA loads exceeding 10%.

Cells were isolated from two groups of 10 tissue samples (n=5 per group), a blank control group (CON), and a specific mode EA stimulation group (EA). The cells were clustered using the Seurat package, and the t-distributed Stochastic Neighbor Embedding algorithm was used for visualization. Twenty-three clusters (cluster 0–cluster 22) were obtained by cell subgroup classification. The proportion of each cluster in the CON and EA groups is shown in *Figure 1c and d*.

### Identification of cell types in different clusters in the CON and EA groups

The SingleR package was used to identify the predominant cell types. Cells were annotated according to the expression of canonical cell class markers (*Figure 2a*). We identified eight cell types (*Figure 2b*). In addition to endothelial cells, we identified astrocytes, microglia, oligodendrocytes, T cells, border-associated macrophages (BAMs), fibroblasts, and granulocytes in this dataset (*Figure 2a and b*). The combined analysis of cell clustering (*Figure 1d*) and cell identification showed that endothelial cells were enriched in two subpopulations, astrocytes were enriched in six subpopulations, and microglial marker genes were enriched in 10 subpopulations. Oligodendrocytes, T cells, BAMs, fibroblasts, and granulocytes each exhibited a single subcluster (*Figure 2b and c*).

### Evaluation of DEGs and pathways related to EA intervention on a holistic level

Based on the differences between the EA and CON groups, the cells with large clustering differences between the groups and those closely related to the composition and function of the BBB (endothelial cells, astrocytes, and microglia) were selected for differential analysis to further analyze the mechanism of EA intervention in the BBB.

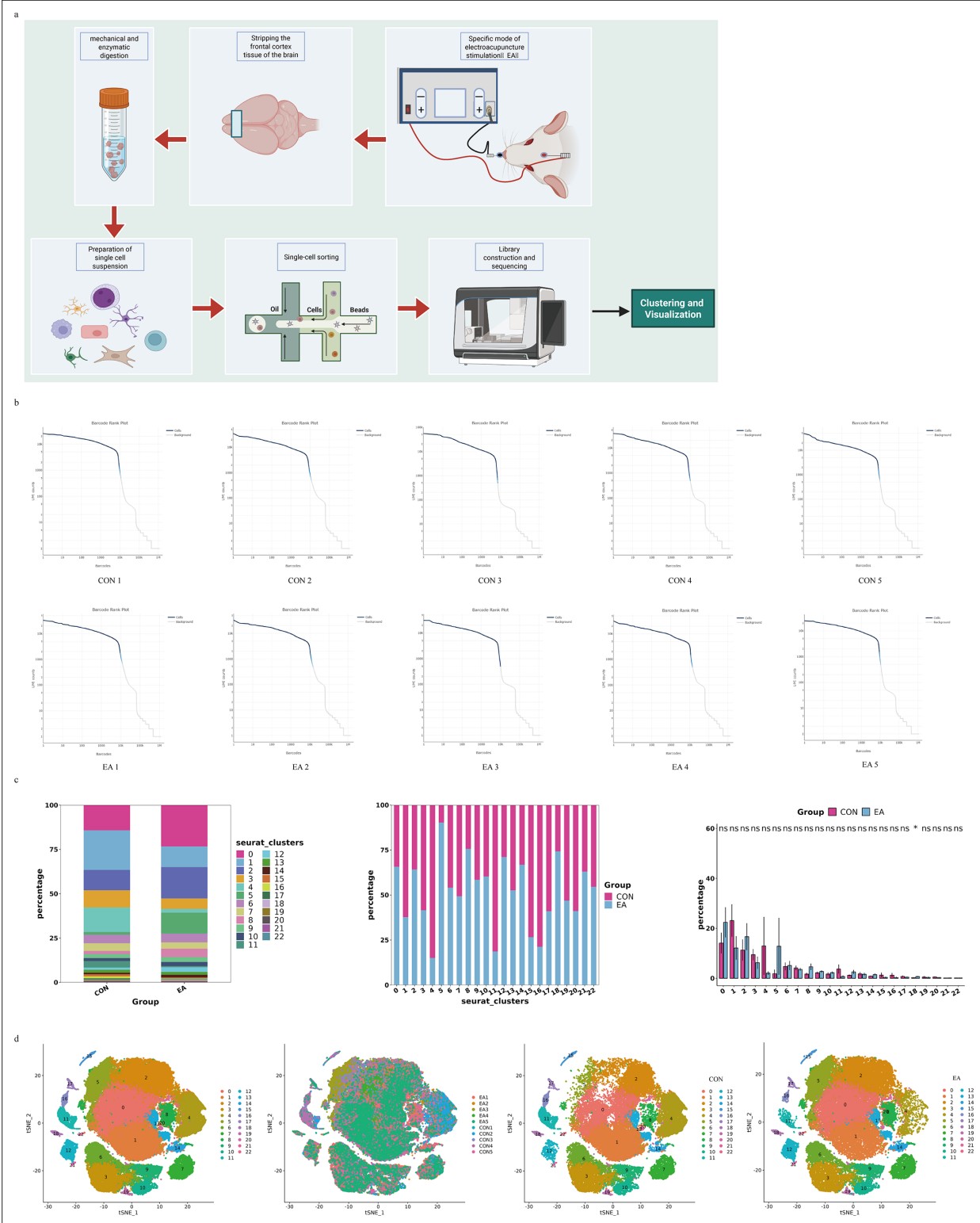

**Figure 1.** Overview of single-cell analysis of the experimental samples. (**a**) Schematic overview of the study design. (**b**) Results of effective cell identification. (**c**) Percentage of 23 clusters in EA and CON groups. (**d**) T-Distributed Stochastic Neighbor Embedding (t-SNE) of the transcriptome from cells of the CON and EA groups. CON, control; EA, electroacupuncture.

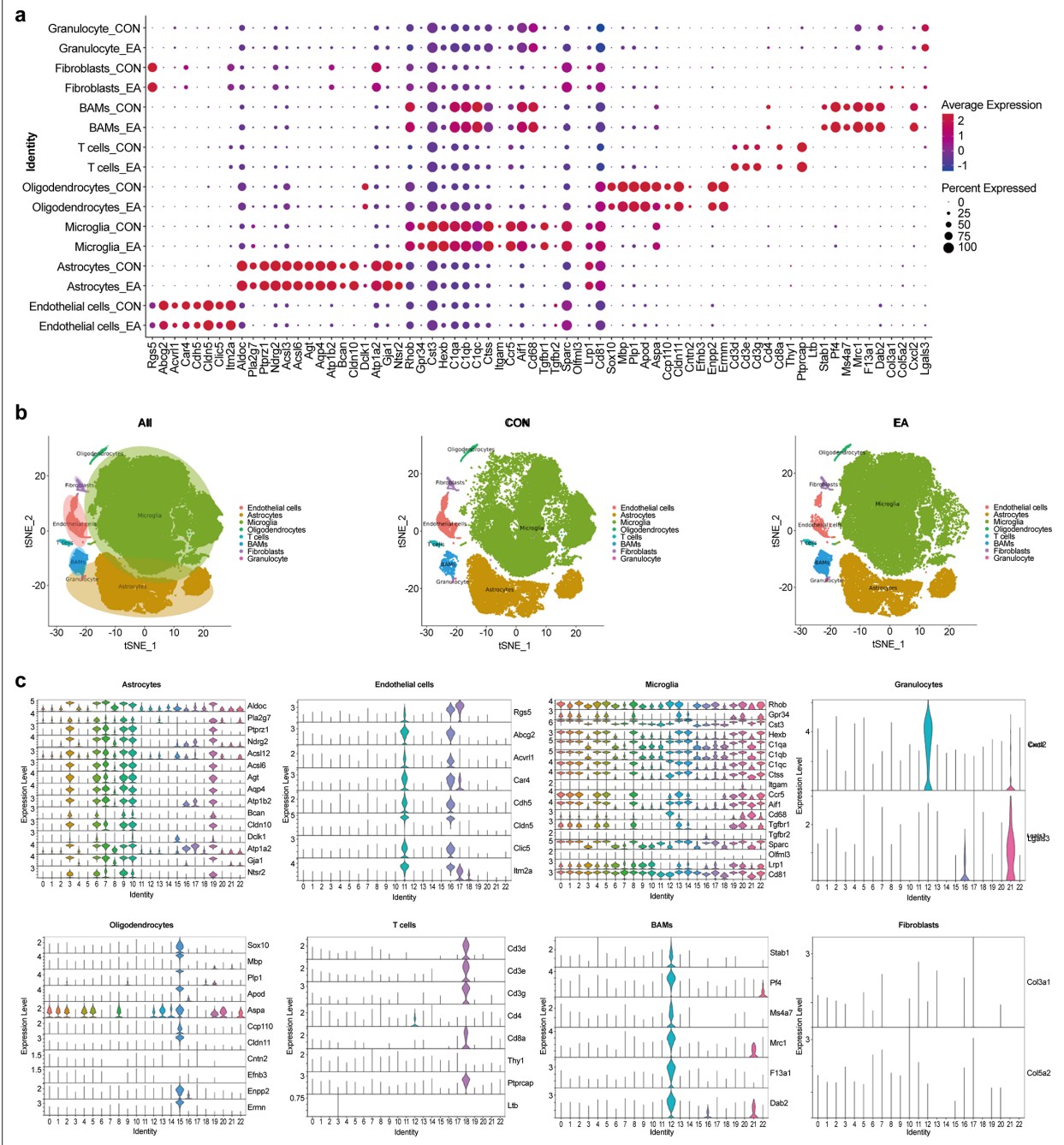

**Figure 2.** Results of the cell type identification. (**a**) Dot plot heatmap of the marker genes in individual clusters. (**b**) Cell type identification T-Distributed Stochastic Neighbor Embedding (t-SNE) diagram. (**c**) The violin plot shows the expression of marker genes of eight cells. CON, control; EA, electroacupuncture.

## Endothelial cells

Endothelial cells play a crucial role in BBB function. These cells create a continuous network of tight and adhesive connections along their contacts, forming a selective barrier that regulates the movement of substances inside and outside the brain. The DEGs were analyzed to identify important genes and pathways involved in maintaining this barrier. The expression of a total of 147 genes was significantly increased, and that of 160 genes was markedly downregulated (logFC = 0.26, p<0.01). The expression fold changes of the top 10 genes most significantly upregulated and downregulated are shown in *Figure 3a*, and all DEGs are listed in *Supplementary file 1*.

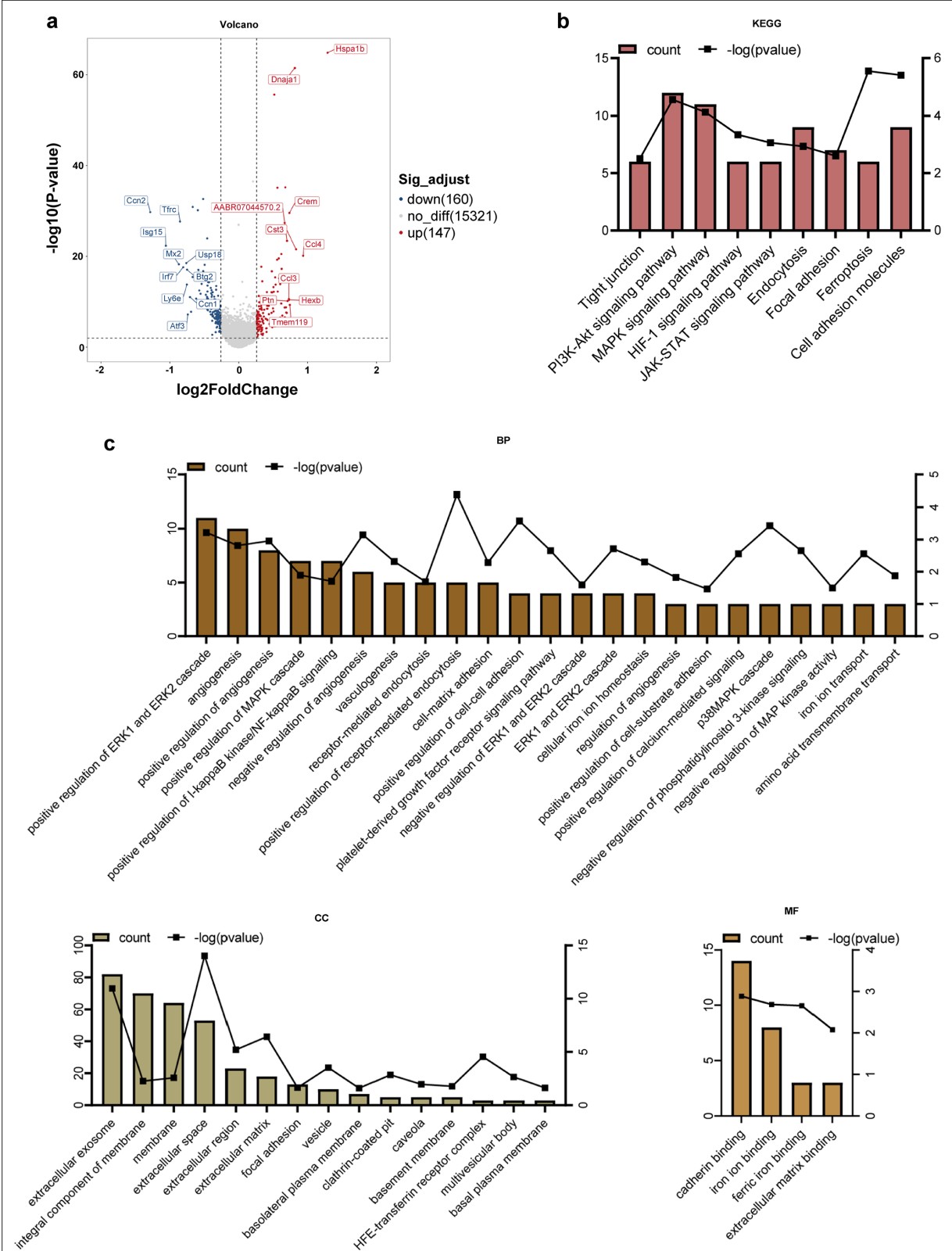

**Figure 3.** Differential gene and enrichment analysis results of the endothelial cells (electroacupuncture, EA vs control group, CON). (**a**) Differential gene volcano map. (**b**) Histogram of the KEGG analysis results. (**c**) Histogram of the GO analysis results. GO, Gene Ontology; KEGG, Kyoto Encyclopedia of Genes and Genomes.

To perform enrichment analysis to evaluate the functionality of the 307 genes, we used the Gene Ontology (GO) terms, which encompassed biological processes (BPs), cellular components (CCs), and molecular function (MF), demonstrating the enrichment of 42 GO terms related to the BBB ($p<0.05$, count $\geq$3) (*Figure 3c*). KEGG analysis yielded nine pathways that may be associated with BBB function: tight junctions, endocytosis, focal adhesion, the phosphatidylinositol 3-kinase (PI3K)/Akt signaling pathway, the mitogen-activated protein kinase (MAPK) signaling pathway, the hypoxia-inducible factor 1 (HIF-1) signaling pathway, the Janus kinase/signal transducer and activator of transcription (JAK/STAT) signaling pathway, ferroptosis, and cell adhesion molecules (*Figure 3b*). Computational gene set enrichment analysis ($q<0.05$) revealed eight significantly enriched unique pathways and identified the following BPs as being affected by EA: complement and coagulation cascades, ribosomes,

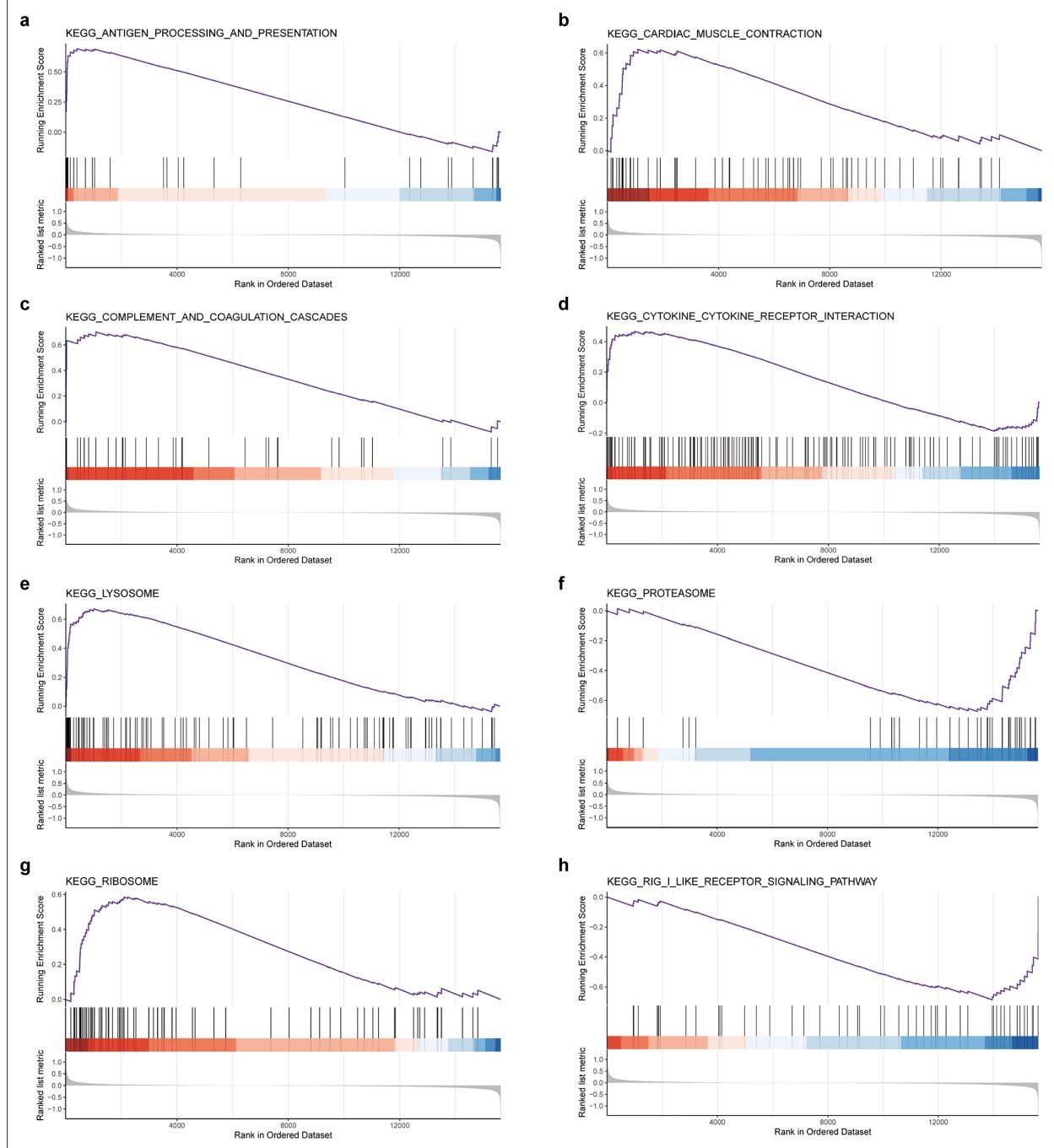

**Figure 4.** Gene set enrichment analysis results (electroacupuncture, EA vs control group, CON). (**a-h**) Eight pathways of GSEA results, EA vs CON.

lysosomes, cardiac muscle contraction, the rig I-like receptor signaling pathway, proteasomes, antigen processing and presentation, and cytokine-cytokine receptor interactions (*Figure 4*).

## DEGs in astrocytes

DEG analysis identified 24 downregulated and three upregulated genes, and the top 10 most upregulated and top three most downregulated genes are shown in *Figure 5a*. GO was performed. KEGG analyses revealed that DEGs were mainly enriched in calcium-related terms, adenosine triphosphate (ATP) and adenosine diphosphate synthesis and metabolism, and the nicotinamide adenine dinucleotide dehydrogenase complex. These molecules are prominent in certain pathways that may be connected to BBB function and permeability, including positive regulation of vascular permeability, cellular response to the vascular endothelial growth factor stimulus, endothelial cell chemotaxis, and the platelet-derived growth factor receptor signaling pathway (*Figure 5b and c*). Furthermore, these molecules were also involved in the regulation of the metabolic process of reactive oxygen species, the response to hypoxia, the response to hyperoxia, cell respiration, and the regulation of gaseous respiratory exchange.

## DEGs in the microglia following EA intervention

Among the DEGs observed in the microglia under EA intervention, the top 10 genes that were most significantly upregulated and downregulated are shown in *Figure 6a*. EA produced an immediate damage-associated molecular pattern response and the co-chaperone of the heat shock protein 70, DNAJ protein family (Dnaja1 and Dnajb1), related to the inflammatory response (*Cao et al., 2014*; *Kovacs et al., 2017*). According to the GO analysis, the terms associated with DEGs were mainly classified into five categories: calcium signaling, MAPK-related pathways, chemokine pathways, extracellular function, and angiogenesis (*Figure 6c*). *Figure 6b* shows the results of functional enrichment in the KEGG pathway database, which may be related to BBB function: the MAPK signaling pathway, endocytosis, HIF-1 signaling pathway, PI3K-Akt signaling pathway, tumor necrosis factor (TNF) signaling pathway, cell adhesion molecules, Wnt signaling pathway, and tight junctions.

## Endothelial cell subsets following EA intervention

### Characteristic gene expression of EC_Clusters

The single-cell analysis identified 23 EC subgroups, of which four clusters (clusters 0, 2, 4, and 5) were dominant between the EA and control groups (*Figure 7a*). Therefore, further GO and KEGG analysis was performed to evaluate the characteristic gene expression patterns of these four subgroups (*Figure 7c–f*), and the bubble maps of the top 10 genes expressed in these four subgroups are shown in *Figure 7b*.

The upregulated and downregulated genes in these four subgroups were further analyzed (EA vs. CON). The enriched pathways of DEGs in EC_Cluster0 and EC_Cluster2 exhibited a high degree of overlap (*Figure 7c and d*) and included infectious disease, the immune system, the MAPK signaling pathway, fluid shear stress, and atherosclerosis. In contrast, cluster 0 was enriched in platelet activation, whereas cluster 2 was enriched in the TNF signaling pathway genes. GO and KEGG results showed that the characteristic expression genes of EC_Cluster4 were mainly involved in immune system-related terms (*Figure 7e*). Similarly, the DEGs of EC_Cluster5 are primarily enriched in pathways related to the immune system, infectious diseases, and the regulation of cell population proliferation. (*Figure 7f*). In contrast to EC_Cluster4, EC_Cluster5 was also enriched in the interleukin (IL)–17 signaling pathway, fluid shear stress, and the atherosclerosis pathway.

### Electroacupuncture altered EC_Clusters of interest

To examine the DEGs and pathways associated with EA exposure from the four subsets of EC, we compared the significantly upregulated or downregulated genes between the EA and CON groups and identified the most representative enriched pathways and processes (*Figures 8 and 9*). Neither the upregulated nor downregulated genes of EC_Cluster0 were independent of other clusters; that is, the differential genes of EC_Cluster0 represented the common influence of EA intervention, suggesting that this is the basic response of endothelial cells to the effect of EA.

Thirty-four genes were significantly upregulated, and 11 genes were significantly downregulated by EA in EC_Cluster0 (*Figure 8b*). Pathway analysis revealed downregulation of cellular responses to

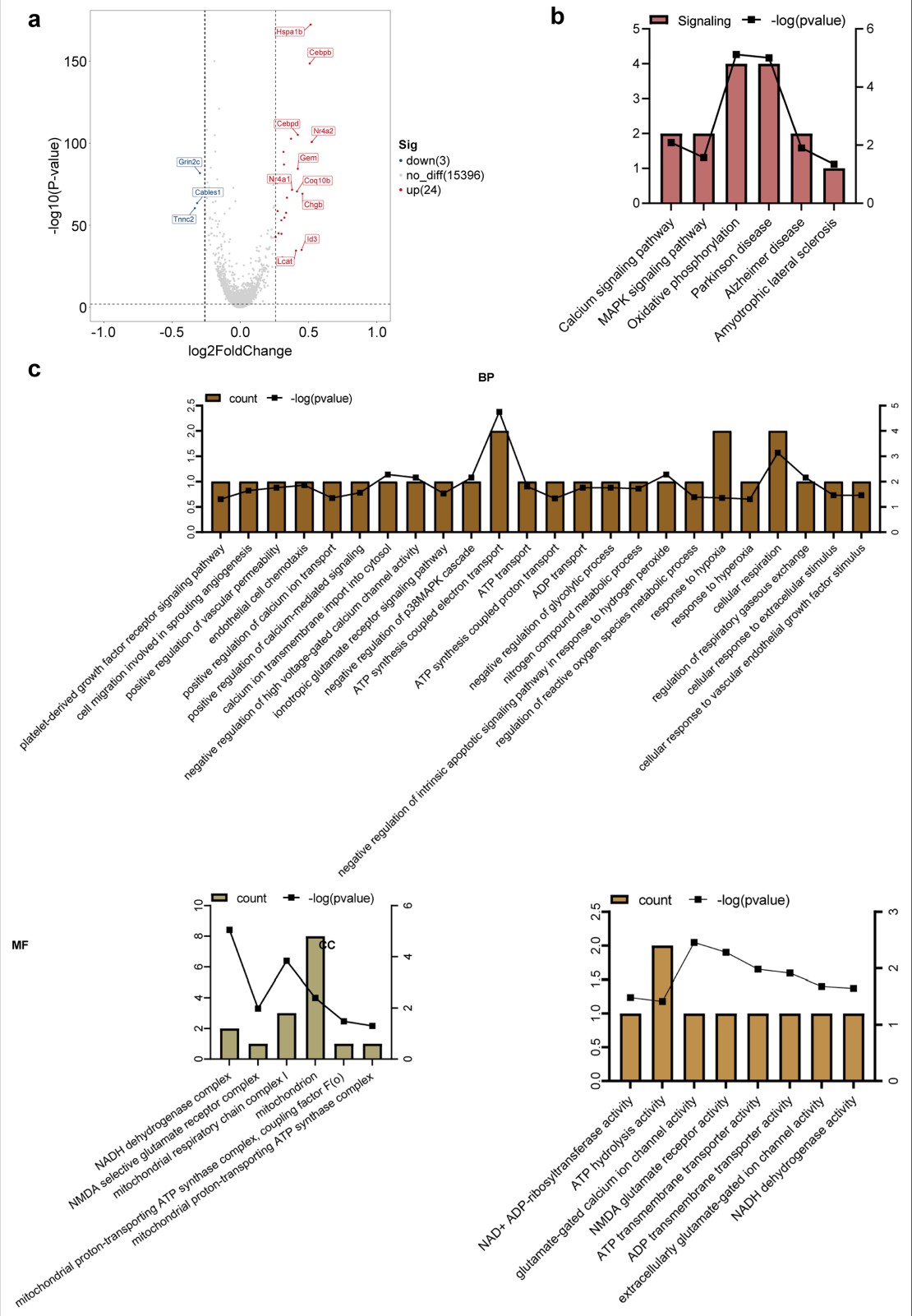

**Figure 5.** Differential gene and enrichment analysis results of astrocytes (electroacupuncture, EA vs control group, CON). (**a**) Differential gene volcano map. (**b**) Histogram of KEGG analysis results. (**c**) Histogram of GO analysis results. GO, Gene Ontology; KEGG, Kyoto Encyclopedia of Genes and Genomes.

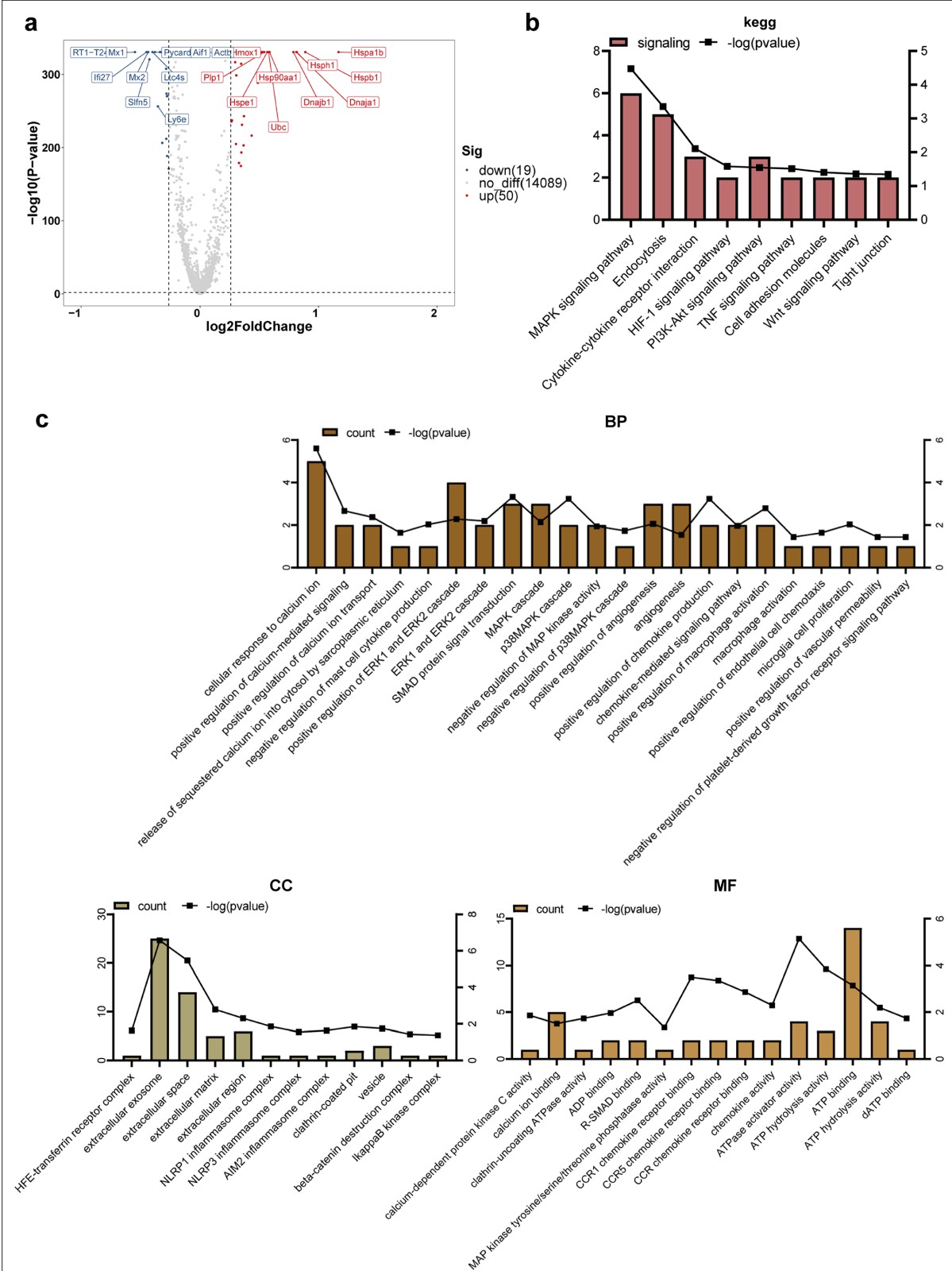

**Figure 6.** Differential gene and enrichment analysis results of the microglia (electroacupuncture, EA vs control group, CON). (**a**) Differential gene volcano map. (**b**) Histogram of KEGG analysis results. (**c**) Histogram of GO analysis results. GO, Gene Ontology; KEGG, Kyoto Encyclopedia of Genes and Genomes.

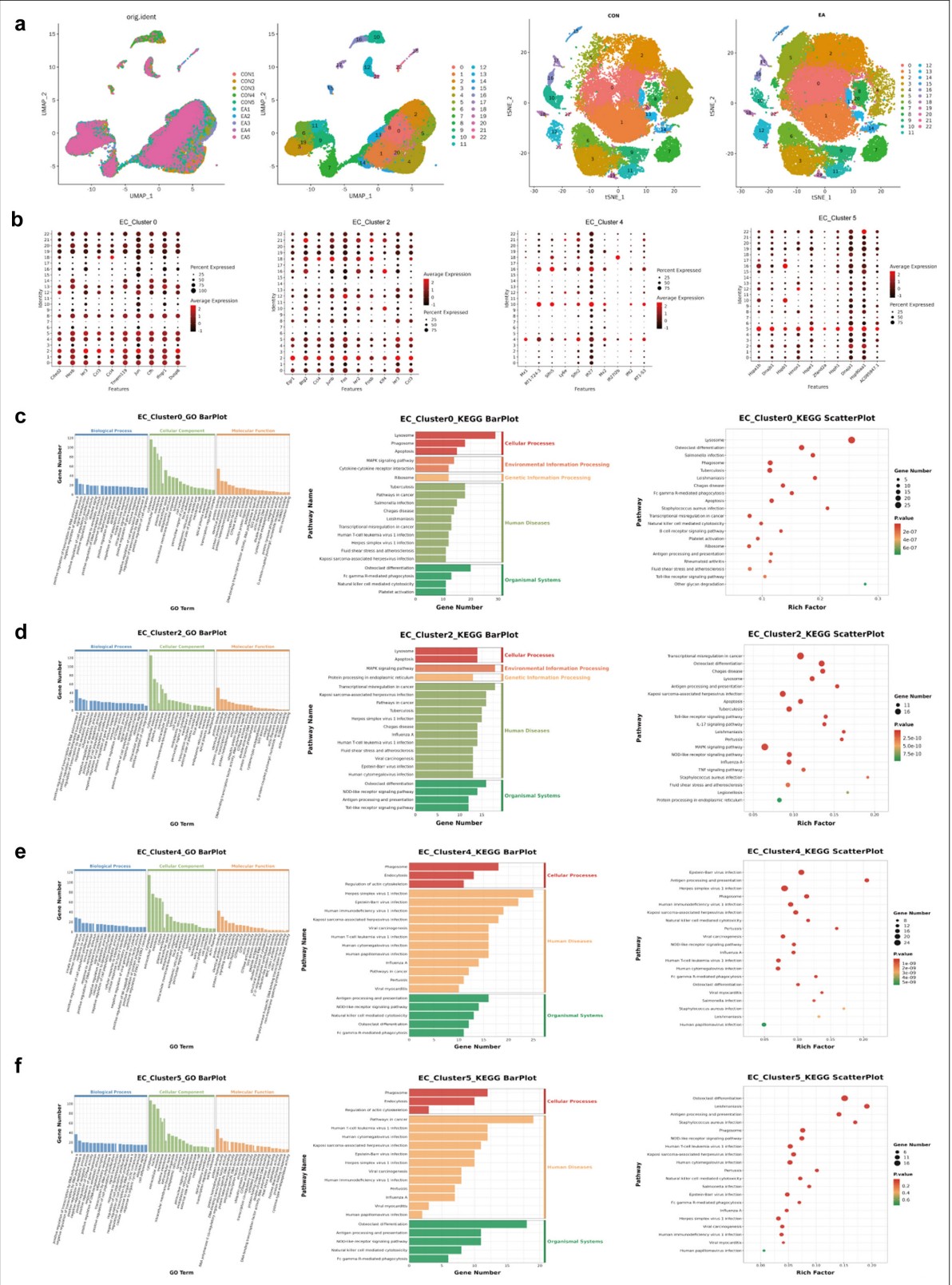

**Figure 7.** Subgroup clustering and functional enrichment of endothelial cells (electroacupuncture, EA vs control group, CON). (**a**) Endothelial cell subset analysis in the two groups. (**b**) The top 10 characteristic expression genes of the four subgroups. (**c–f**) The GO and KEGG results of the characteristic expression genes of the four subgroups. The histogram of GO results show that the number of enriched genes was ranked in the top 20, and *q*<0.05. The histogram and bubble diagram of the KEGG results show the top 20 terms of the number of enriched genes and the 20 terms with the smallest p-value, respectively. GO, Gene Ontology; KEGG, Kyoto Encyclopedia of Genes and Genomes.

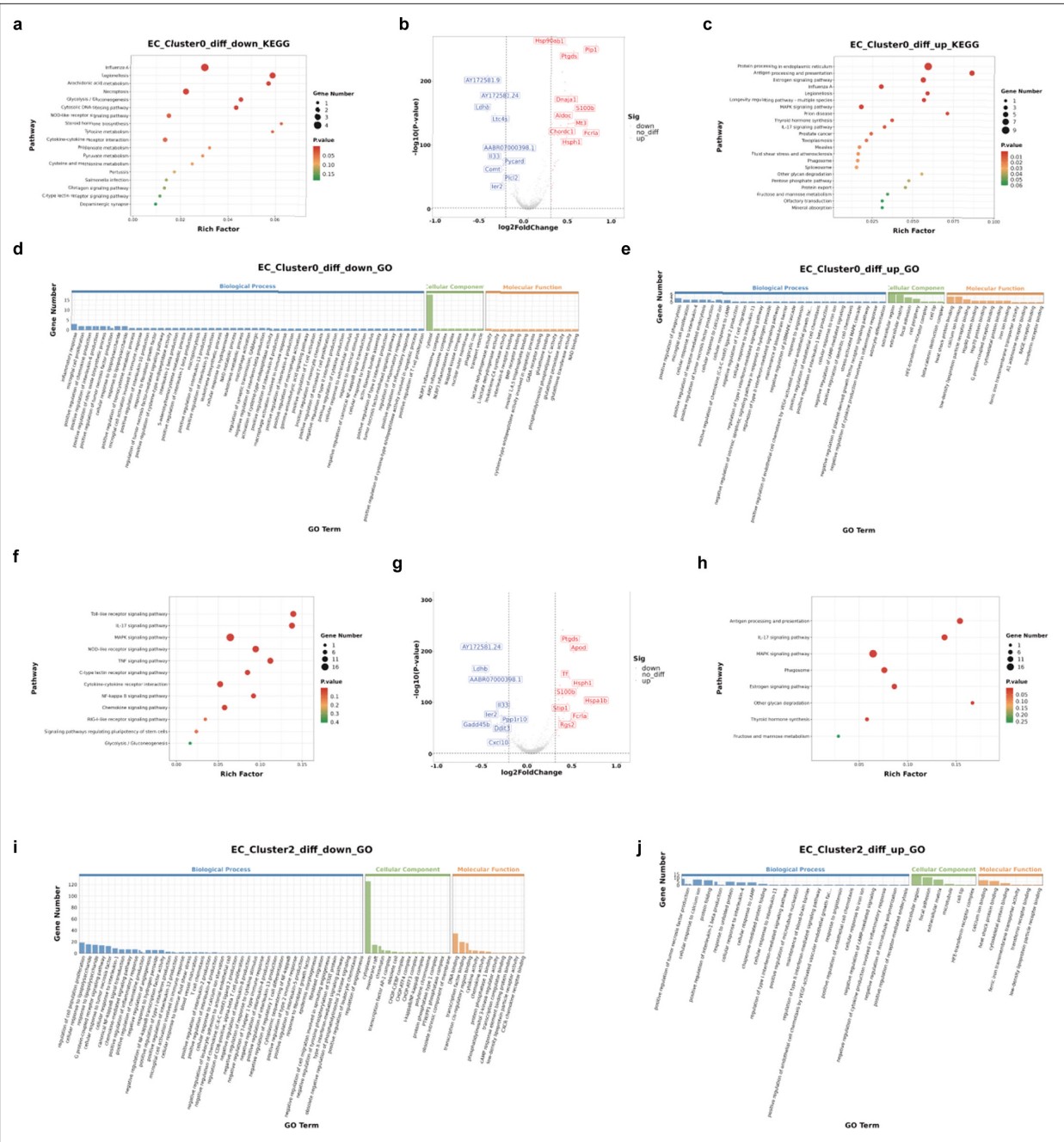

**Figure 8.** Differential gene and enrichment analysis results of EC_Cluster0 and EC_Cluster2. (**a**) KEGG enrichment results for down-regulated genes in EC_Cluster0. (**b**) EC_Cluster0 differential gene volcano map. (**c**) KEGG results of EC_Cluster0 upregulated genes. (**d**) GO results of EC_Cluster 0 downregulated genes. (**e**) GO results of EC_Cluster0 upregulated genes. (**f**) KEGG results of EC_Cluster2 downregulated genes. (**g**) EC_Cluster2 differential gene volcano map. (**h**) KEGG results of EC_Cluster2 upregulated genes. (**i**) GO results of EC_Cluster2 downregulated genes. (**j**) GO results of EC_Cluster2 upregulated genes. GO, Gene Ontology; KEGG, Kyoto Encyclopedia of Genes and Genomes.

extracellular stimuli and inflammatory responses (*Supplementary file 1*, *Figure 8a and d*). Genes in pathways that involve leukotriene biosynthetic processes, metabolic processes, and synthase activity were downregulated (*Supplementary file 1*). We found that EA-related genes related to interferon signaling were upregulated by EA (*Figure 8b* and *Supplementary file 2*), which affects brain endo-thelial function and barrier integrity (*Oshima et al., 2001*; *Zhang et al., 2005*; *Mbofung et al., 2017*). Interestingly, Stip1 regulated the Hsp90ab1 function (*Li et al., 2014*), and its mRNA was also regulated

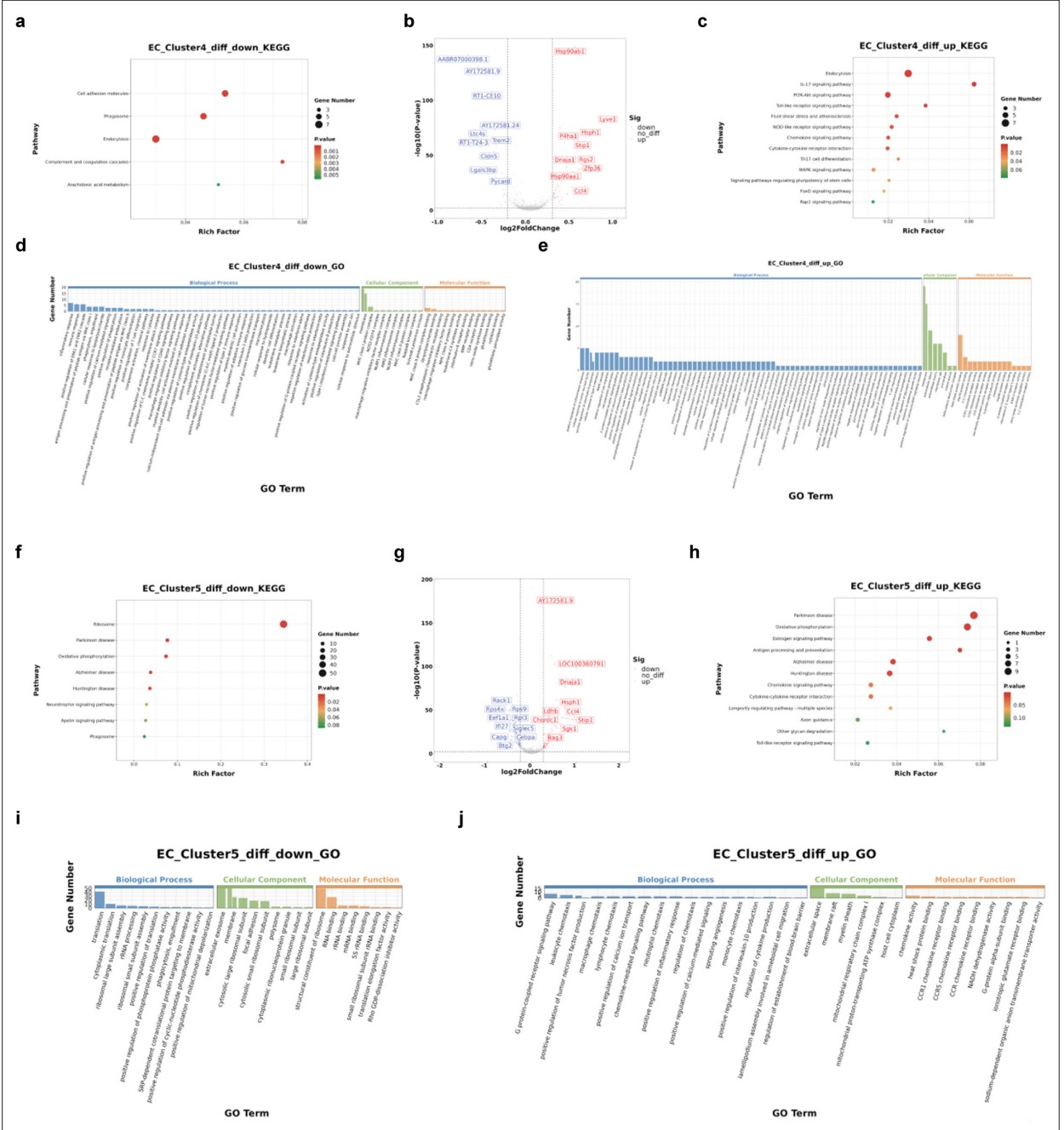

**Figure 9.** Differential gene and enrichment analysis results of EC_Cluster4 and EC_Cluster5. (**a**) KEGG results of EC_Cluster4 downregulated genes. (**b**) EC_Cluster4 differential gene volcano map. (**c**) KEGG results of EC_Cluster4 upregulated genes. (**d**) GO results of EC_Cluster4 downregulated genes. (**e**) GO results of EC_Cluster4 upregulated genes. (**f**) KEGG results of EC_Cluster5 downregulated genes. (**g**) EC_Cluster5 differential gene volcano map. (**h**) KEGG results of EC_Cluster5 upregulated genes. (**i**) GO results of EC_Cluster5 downregulated genes. (**j**) GO results of EC_Cluster5 upregulated genes. GO, Gene Ontology; KEGG, Kyoto Encyclopedia of Genes and Genomes.

by EA in EC_Cluster0 (*Supplementary file 2*). Notably, *Hsp90b1* and *Hsp90ab1* were enriched in the fluid shear stress and atherosclerosis pathways (*Figure 8c*).

The upregulated genes and corresponding functions of EC_Cluster2 were highly correlated with those of EC_Cluster0, with the addition of increased *Psap, Mt-co3, Pdia4,* and *Atrx.1* expression (*Figure 8g, h, j* and *Figure 10a*) related to the extracellular matrix region and active transmembrane transporter activity (*Supplementary file 3*). Only 20 genes were downregulated in EC_Cluster2 (*Figure 10b*). Among them, *Nfkbia, Cxcl10,* and *Cxcl2* had the highest number of enriched pathways,

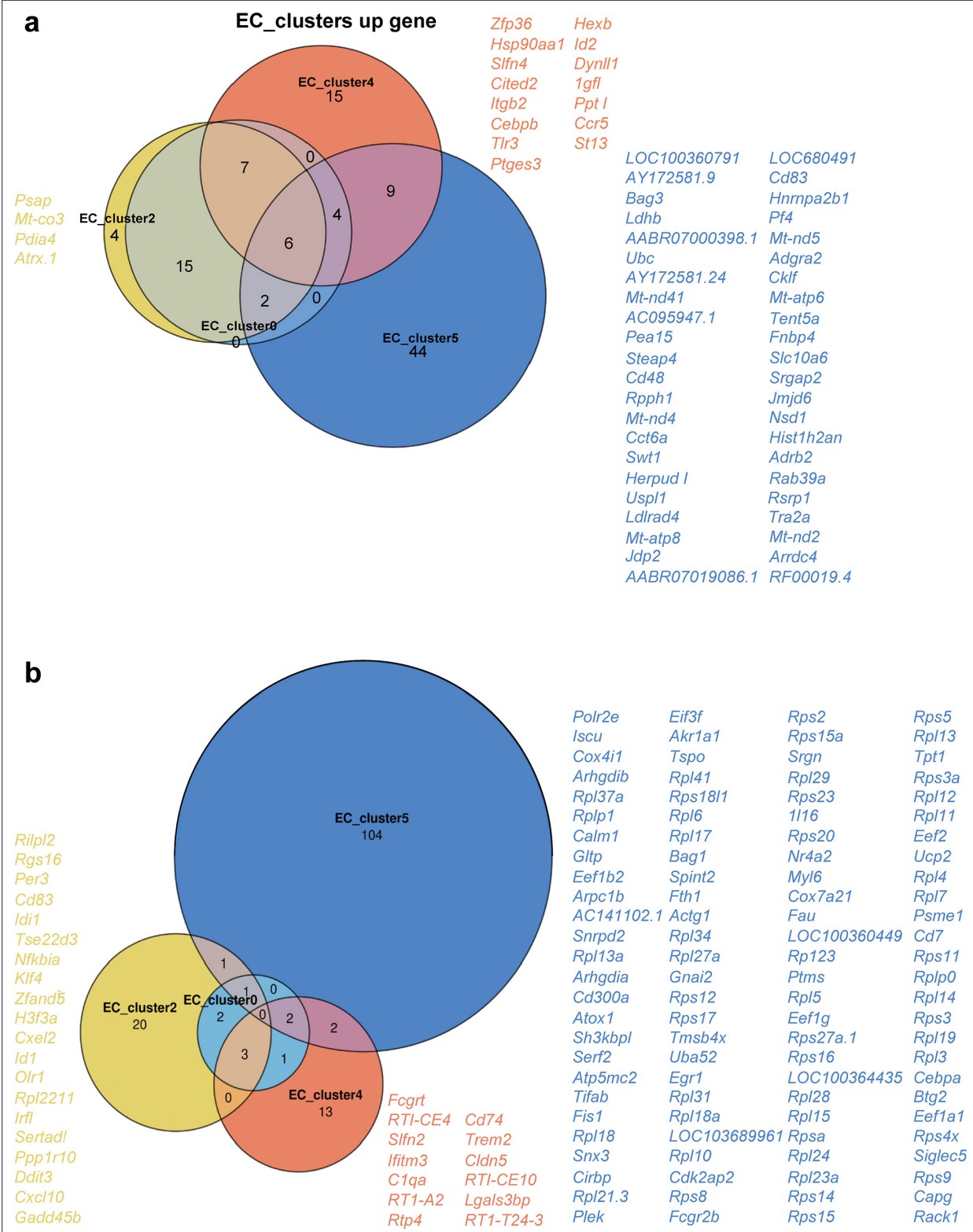

**Figure 10.** Venn diagram showing the differential gene intersection and uniqueness of the four clusters of interest in endothelial cells. (**a**) Intersection and uniqueness of upregulated genes among the four clusters of interest in endothelial cells. (**b**) Intersection and uniqueness of downregulated genes among the four clusters of interest in endothelial cells.

mainly including the NF-κB signaling pathway, IL-17 signaling pathway, TNF signaling pathway, cytokine-cytokine receptor interaction, and chemokine signaling pathway (*Figure 8f, g and i*, and *Supplementary file 4*).

After EA intervention, 15 unique upregulated genes and 13 downregulated genes were identified in EC_Cluster4 (*Figure 10*). Importantly, downregulated gene enrichment analysis identified pathways related to BBB transport and barrier function. We also observed that terms and pathways related to the immune response were enriched (*Figure 9a, b and d*, and *Supplementary file 5*). The upregulated DEGs were mainly related to the inflammatory response and involved a variety of cytokine-related signaling pathways. Simultaneously, several typical signaling pathways were also observed, including the ERK cascade, MAPK, and PI3K pathways. Finally, the EA intervention in EC_Cluster4 also resulted in changes in vasculogenesis, astrocyte cell migration, pinocytosis, and calcium-mediated signaling (*Figure 9b, c and e*, and *Supplementary file 6*).

Among the downregulated genes in EC_Cluster5, only six genes overlapped with other subgroups, and the remaining 104 genes were downregulated (*Figure 10b*). The downregulated genes were primarily related to ribosomes, suggesting active changes in protein synthesis and processing (*Supplementary file 7*). Among the upregulated genes in EC_Cluster 5, 21 coincided with other subgroups, and the remaining 44 genes (*Figure 10a*) were enriched in the regulation of the establishment of the BBB, sodium-dependent organic anion transmembrane transporter activity, and sprouting, which are important for BBB formation and material transport, in addition to chemokine activity and immune cell chemotaxis (*Figure 9g, h and j*, and *Supplementary file 8*).

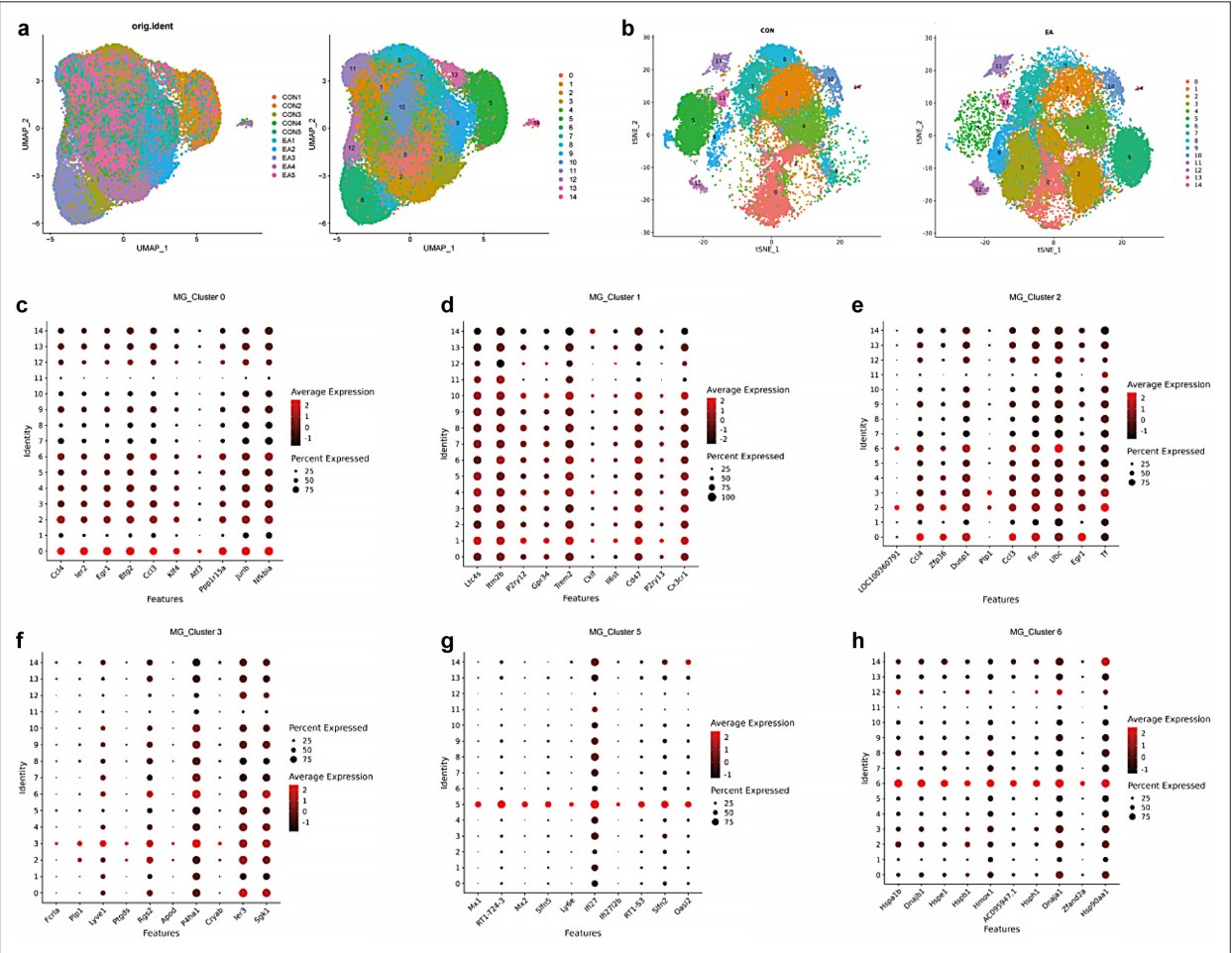

**Figure 11.** Subgroup clustering and functional enrichment of microglia. (**a**, **b**) Microglia subset analysis in the two groups. (**c–h**) The top 10 characteristic expression genes of the six clusters of interest.

### Nature of the microglial cell subsets and the effects of the EA intervention

Analysis of the microglial cell subsets yielded 15 cluster subsets, and marker genes showed different microglial subsets (*Figure 11*); therefore, these 15 clusters were considered distinct clusters of microglial cells. We selected six clusters (MG_Cluster0, 1, 2, 3, 5, and 6) with large changes and annotated marker genes that were differentially expressed among these cell populations. GO annotation identified the top 20 enrichment terms with the most significant enrichment, whereas KEGG revealed the top 20 pathways with the largest number of genes and the 20 pathways with the smallest p-values. A Venn diagram displays the overlap between these pathways and entries (*Figures 12 and 13*).

### Characteristic DEGs of MG_Clusters

The BP, CC, and MF pathways with the most significant (minimum Q-value) enrichment of the top gene of MG_Cluster1, as well as the top 20 pathways with the highest number of these genes, are displayed in *Figures 12b and 13b*, and *Supplementary files 9–11*. These entries involved the membrane dynamics, G protein-coupled receptor signaling pathway, cytokine-mediated signaling pathway, and extracellular space (*Figures 12 and 13a*, and *Supplementary file 9*). KEGG analysis showed cytokine-cytokine receptor interaction, arachidonic acid metabolism, ECM-receptor interaction, signaling pathways regulating stem cell pluripotency, platelet activation, and JAK-STAT signaling (*Figures 12b, 13b and c*, *Supplementary file 10* and *Supplementary file 11*). The uniquely enriched entries and pathways of MG_Cluster0 were mainly related to the intracellular anatomical structure, responses of cells to various cytokines, and regulation of transcription factors (*Figures 12a and 13a*, and *Supplementary file 12*). The important pathways were the FoxO and NF-κB signaling pathways (*Figures 12a, 13b and c*, *Supplementary file 12* and *Supplementary file 13*). In contrast to the other clusters, the functions of MG_Cluster2 included the extracellular region, cellular response to organic substances, and lysosome and chemokine signaling pathways (*Figures 12c and 13*, *Supplementary files 14–17*). MG_Cluster3 had important functions that differed from those of the other clusters, summarized as negative regulation of amyloid fibril formation, positive regulation of tau protein kinase activity, and low-density lipoprotein particle receptor binding (*Figures 12d and 13a*, and *Supplementary file 18*). The total number of entries and pathways representing the unique functions of MG_Cluster5 was the largest (*Figure 13*), and its functions mainly focused on the immune response (*Figures 12e and 13a–c*, *Supplementary files 19–21*). For MG_Cluster6, the top enriched pathways in MG_Cluster6 were not unique to other clusters (*Figure 13b and c*, *Supplementary files 21–23*), and GO enrichment analysis showed that they were mainly related to ATP binding, transcription, and protein folding (*Figures 12f, 13b and c*, and *Supplementary file 22*).

## DEGs following electroacupuncture intervention in the MG_Clusters

We analyzed the coincidence and non-coincidence of the upregulated and downregulated DEGs in the six clusters (*Figures 14a and 15a*). Similarly, we performed an intersection analysis of GO entries with more than two DEGs and KEGG pathways with a Q-value ≤0.05 (*Figures 14b, 15b and I*).

## Upregulated genes and enrichment analysis of MG_Clusters

Regarding the quantity of upregulated genes, MG_Cluster3 had the highest number (n=52), followed by MG_Cluster6 (n=38); the numbers in MG_Cluster5 and MG_Cluster0 were 14 and 13, respectively; the number in MG_Cluster2 was seven, while the number in MG_Cluster1 was only one (*Figure 14a*). In contrast, GO analysis showed that MG_Cluster5 had the largest number of entries, independent of the other clusters (n=53), followed by MG_Cluster6 and MG_Cluster0, with 18 and 13 entries, respectively. MG_Cluster2, MG_Cluster1, and MG_Cluster3 had similar numbers of independent enrichment entries; MG_Cluster2 had eight entries, and the remaining two clusters had seven entries (*Figure 14b*). Regarding the KEGG pathways, for MG_Cluster0, all activated pathways were similar to those of the other clusters. In contrast, the number of upregulated gene enrichment pathways in MG_Cluster1 that do not overlap with other clusters was the largest (13 pathways). The enrichment of upregulated genes in MG_Cluster6 occurred in six pathways, whereas the featured pathways in MG_Cluster5, 3, and 2 were 3, 2, and 1, respectively (*Figure 14i*).

Next, based on the above results, we analyzed the unique GO and KEGG enrichment of the six clusters of interest to study the functional changes in each cluster after the EA intervention. Among them, MG_Cluster0 was related to vascular function. MG_Cluster0 functioned as a focal adhesion and

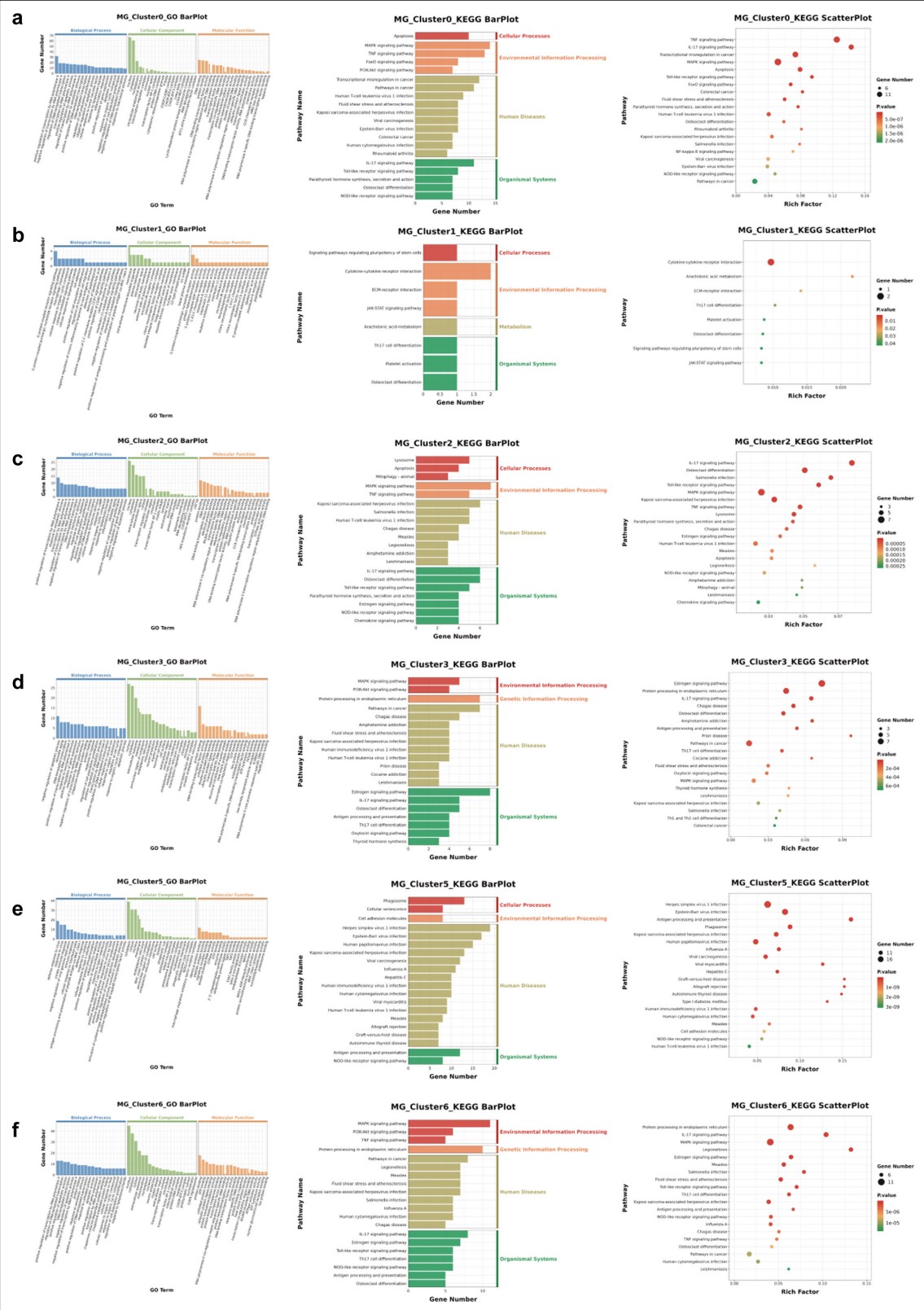

**Figure 12.** The GO and KEGG results of the characteristic expression genes of the six subgroups of microglia. (**a**–**f**) represent the enrichment analysis results of MG_Cluster0, MG_Cluster1, MG_Cluster2, MG_Cluster3, MG_Cluster5, and MG_Cluster6, respectively. The histogram of GO results shows that the number of enriched genes was ranked in the top 20, and q<0.05. The histogram and bubble diagram of the KEGG results show the top 20 terms of the number of enriched genes and the 20 terms with the smallest p-value, respectively. GO, Gene Ontology; KEGG, Kyoto Encyclopedia of Genes and Genomes.

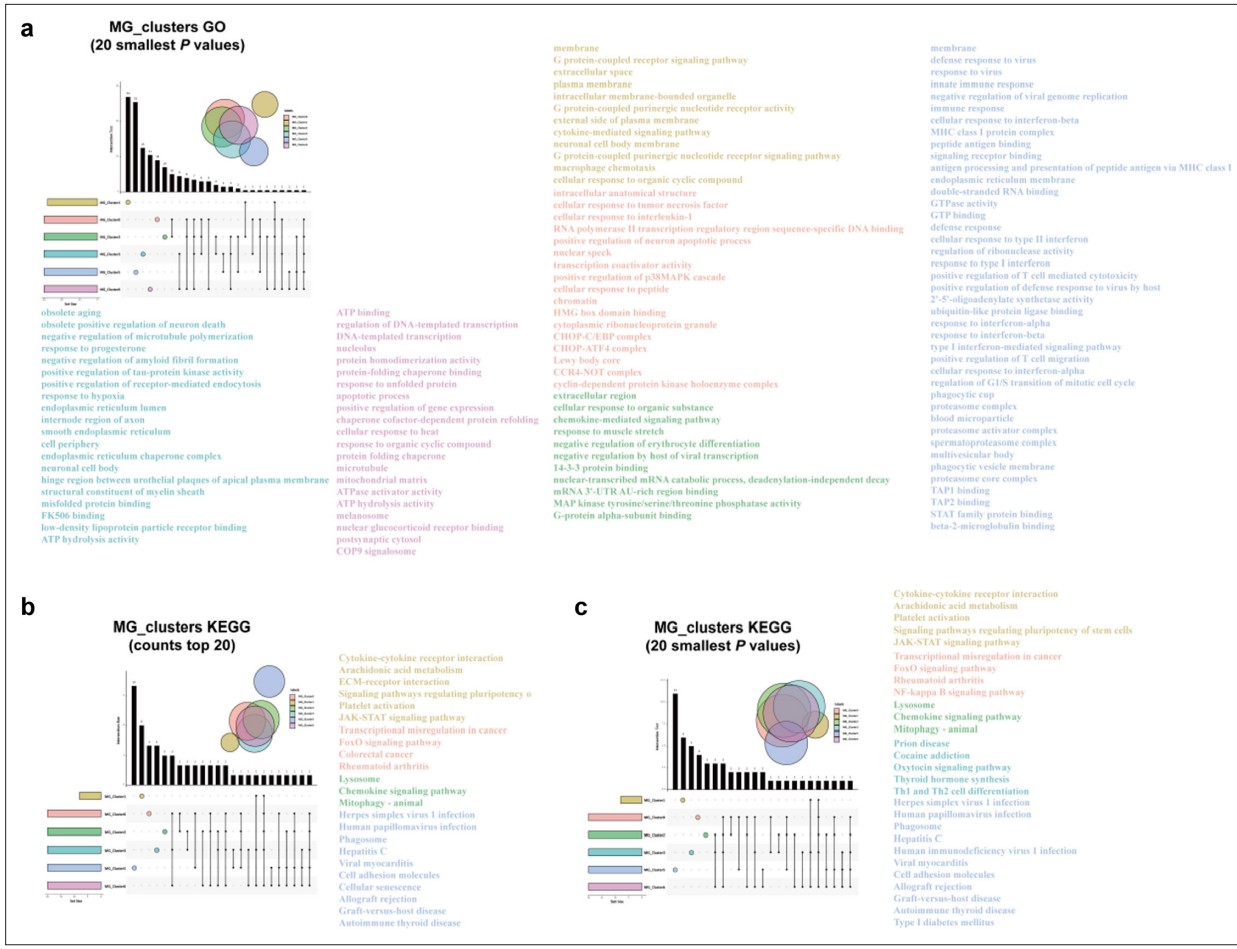

**Figure 13.** Upset map of crossover and uniqueness of the enrichment function of the characteristic genes of 6 MG_Clusters of interest. (**a**) GO results. (**b, c**) KEGG results with the largest number of genes and the most significant enrichment, respectively. GO, Gene Ontology; KEGG, Kyoto Encyclopedia of Genes and Genomes.

synapse (*Figure 14c*). GO enrichment in MG_Cluster1 was mainly related to post-translational modifications of proteins. The KEGG results showed that Hsp90aa1 and Hsp90ab1 were also enriched in the PI3K/Akt signaling pathway (*Figure 14d and j*). In MG_Cluster2, we identified genes upregulated in the cluster after EA intervention, which involved the basement membrane and extracellular region (*Figure 14e and j*). Upregulated genes in MG_Cluster3 were mainly enriched in GO terms related to axon and myelin. KEGG analysis revealed that this cluster was related to ribosomes (*Figure 14f and j*). In MG_Cluster5, the upregulated genes were mainly enriched in the cytosol and plasma membrane. Notably, the two genes upregulated in the cluster (*Ccl3* and *Ccl4*) play key roles in and are mainly enriched in calcium and CCR chemokine receptor-related functions, as well as in the toll-like receptor signaling pathway (*Figure 14g and k*). The enriched pathways in MG_Cluster6 are mostly related to mitochondria after EA intervention (*Figure 14h and k*).

## Downregulated genes and enrichment analysis of MG_Clusters

We observed that among the six clusters of independently downregulated genes, MG_Cluster3 and MG_Cluster6 involved far more genes than the other clusters, with 82 and 68 genes, respectively. The number of pathways identified in the GO and KEGG enrichments of MG_Cluster6 was also higher than those of the other five clusters (*Figure 15a, b and l*), including the cellular response to various stimuli. Another noteworthy pathway was the fluid shear stress and atherosclerosis pathway (*Figure 15h and k*), because the fluid shear stress generated by directional shear of blood flow directly acts on brain microvascular endothelial cells.

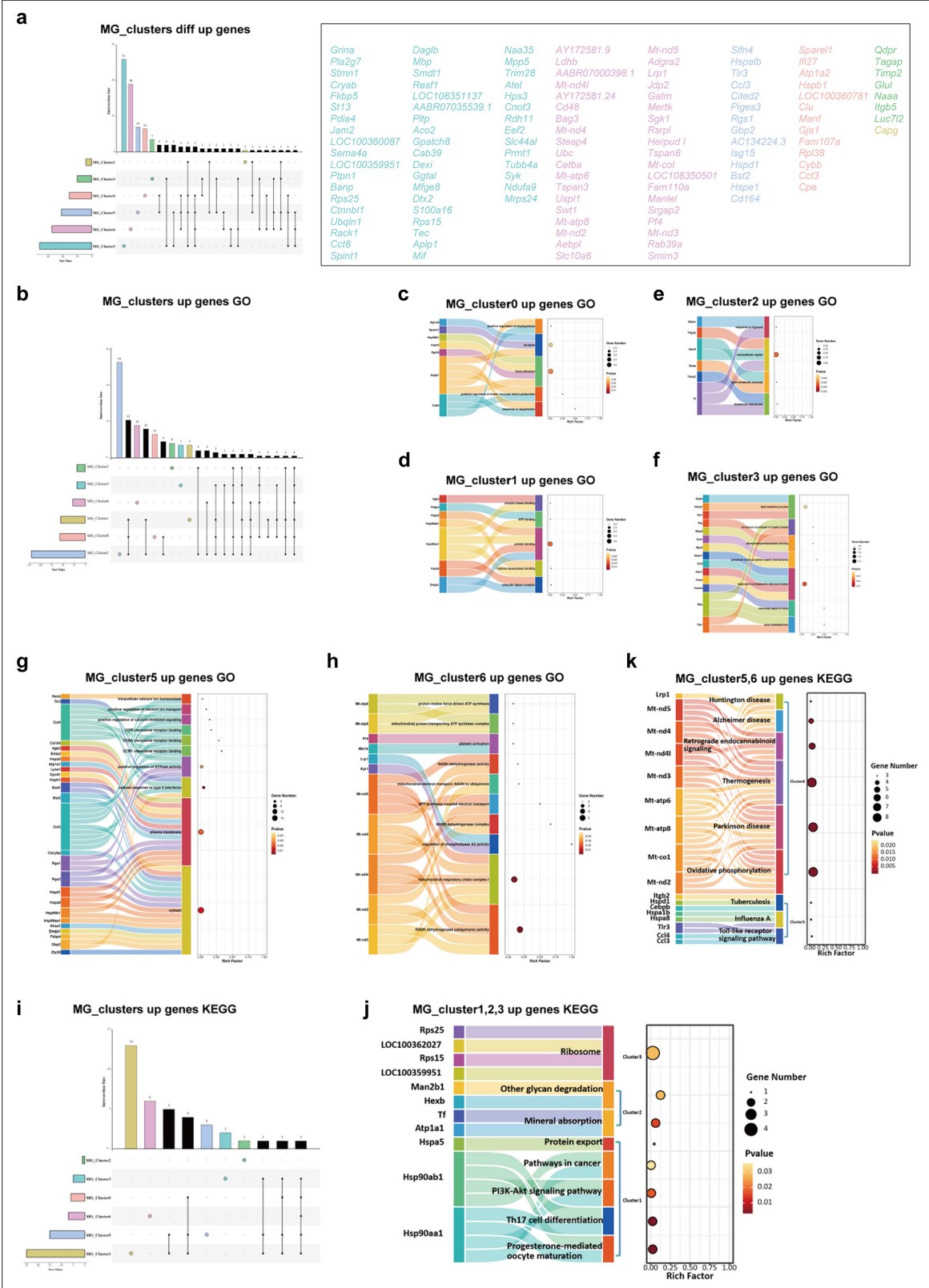

**Figure 14.** Functional enrichment analysis of unique upregulated genes in MG_clusters after electroacupuncture. (**a**) Identical and different upregulated genes in six subpopulations of microglia. The histograms of different colors represent different clusters and gene numbers. Colored circles represent genes that are only upregulated in the corresponding clusters, and these genes are listed in their respective colors. (**b, i**) Crossover and separation of upregulated genes GO and KEGG in six clusters of microglia. (**c–h**) Unique important GO entries and their genes in the six clusters are displayed. (**j, k**)

Figure 14 continued

Unique key KEGG pathways and their genes in the five clusters (MG_Cluster0 has no unique enrichment pathway). GO, Gene Ontology; KEGG, Kyoto Encyclopedia of Genes and Genomes.

Furthermore, the downregulated genes in MG_Cluster0 were mainly associated with focal adhesion, synapses, the response to angiotensin, and positive regulation of angiogenesis; however, no unique KEGG enrichment pathway was identified (*Figure 15c*). The G protein-coupled purine nucleotide receptor and platelet activation were the key terms of MG_Cluster1 enriched (*Figure 15d and j*). In the GO enrichment of MG_Cluster2, nine important genes were downregulated (*Figure 15g and l*). In the unique GO of MG_Cluster3, we focused on the regulation of T cell and B cell migration. KEGG analysis yielded three pathways, including inositol phosphate metabolism, the C-type lectin receptor signaling pathway, and the phosphatidylinositol signaling system (*Figure 15e and l*). Finally, the MG_Cluster5 downregulated genes enriched GO terms and pathways were mainly related to immune function (*Figure 15f and l*).

## Coordination of cell communication in the BBB following EA intervention

Because the BBB is orchestrated by the coordinated actions and interactions of several different cell types, cell communication analysis is necessary. First, we compared cell communication between the EA and CON groups (*Figure 16a–f*) and further investigated the communication network between the two groups of cells and other cells (*Figure 16g and h*). The number of protein interactions between the different cell types (p<0.05) was calculated to generate a heat map (*Figure 16a and d*), and the heat map values were log (natural logarithm) log-transformed (*Figure 16e and f*). Since endothelial cells are the most direct cell type constituting the physical characteristics of the BBB, we focused on the degree of communication between various types of cells and endothelial cells. Upon comparing the EA and CON groups, we found that BAMs, microglia (MG), T cells, granulocytes, and endothelial cells demonstrated obvious differences in their interactions with endothelial cells (*Figure 16a and d*). The results suggested that the cells with the most active relationship with endothelial cells were astrocytes, fibroblasts, and endothelial cells, whereas the largest changes between the two groups were granulocytes (GC) (*Figure 16b, e, g and h*).

To further explore the interaction between cell types, we analyzed the ligand-receptor pairs of other cells interacting with endothelial cells (*Figure 17a*). Among the top 20 ligand-receptor pairs in the mean value, there were four pairs of interactions with endothelial cells that were different between the groups, namely CD74_APP, PLXNB2_PTN, LAMP1_FAM3C, and CD74_COPA. TNFRSF21_APP was present only in the CON group, whereas CSF1R_CSF1 was specifically present in the EA group (*Figure 17a*). In the CON group, the ligand-receptor pair CD74_APP was not observed, suggesting that the EA intervention activated the ligand-receptor interaction of CD74_APP. In the communication between the AC and the EC, the ligand-receptor pair activated by the EA also contained TNFSF12_TNFRSF25, while GDF11_TGFR_AVR2B was blocked. Additionally, the mean interactions of TYRO3_GAS6, TNFSF13_TNFRSF1A, NOTCH1_JAG2, FGFR2_EPHA4, and FGFR2_PTPRR were consistently decreased in the EA group (*Figure 17b*). The FB-EC communication attracted our attention because there were more ligand-receptor pairs than others. In the EA group, the communication between TGFB3 in the FB and its corresponding receptors in the endothelial cells was activated. Furthermore, apart from the higher mean PLXNB2_PTN in the EA group than that in the CON group, we observed that all other ligand-receptor pairs with mean changes were higher in the CON group (*Figure 17c*).

Except for the TNFSF12_TNFRSF25 exclusively in the EA group and PLXNB2_SEMA4C in the CON group, in the ligand-receptor interaction between microglia and EC, we observed nine pairs of ligand-receptor pairs promoted by EA (*Figure 17d*). In terms of interaction between granulocytes and endothelial cells (GC-EC), many ligand-receptor pairs were present only in the EA or CON group (*Figure 17e*). We then focused on oligodendrocyte and endothelial cell (OC-EC) communication and found that the *p*-values of most ligand-receptor pairs decreased in the EA group, suggesting that EA rendered these interactions more pronounced. In addition, there were also communications that appeared exclusively in one group (*Figure 17f*). In terms of the TC and EC communication, TGFB1_TGFbeta_receptor1 and TGFB1_TGFbeta_receptor2 appeared only in the EA group, whereas AR_Testosterone_byHSD17B12, COL11A2_integrin_a1b1_complex, IFNG_Type_II_IFNR,

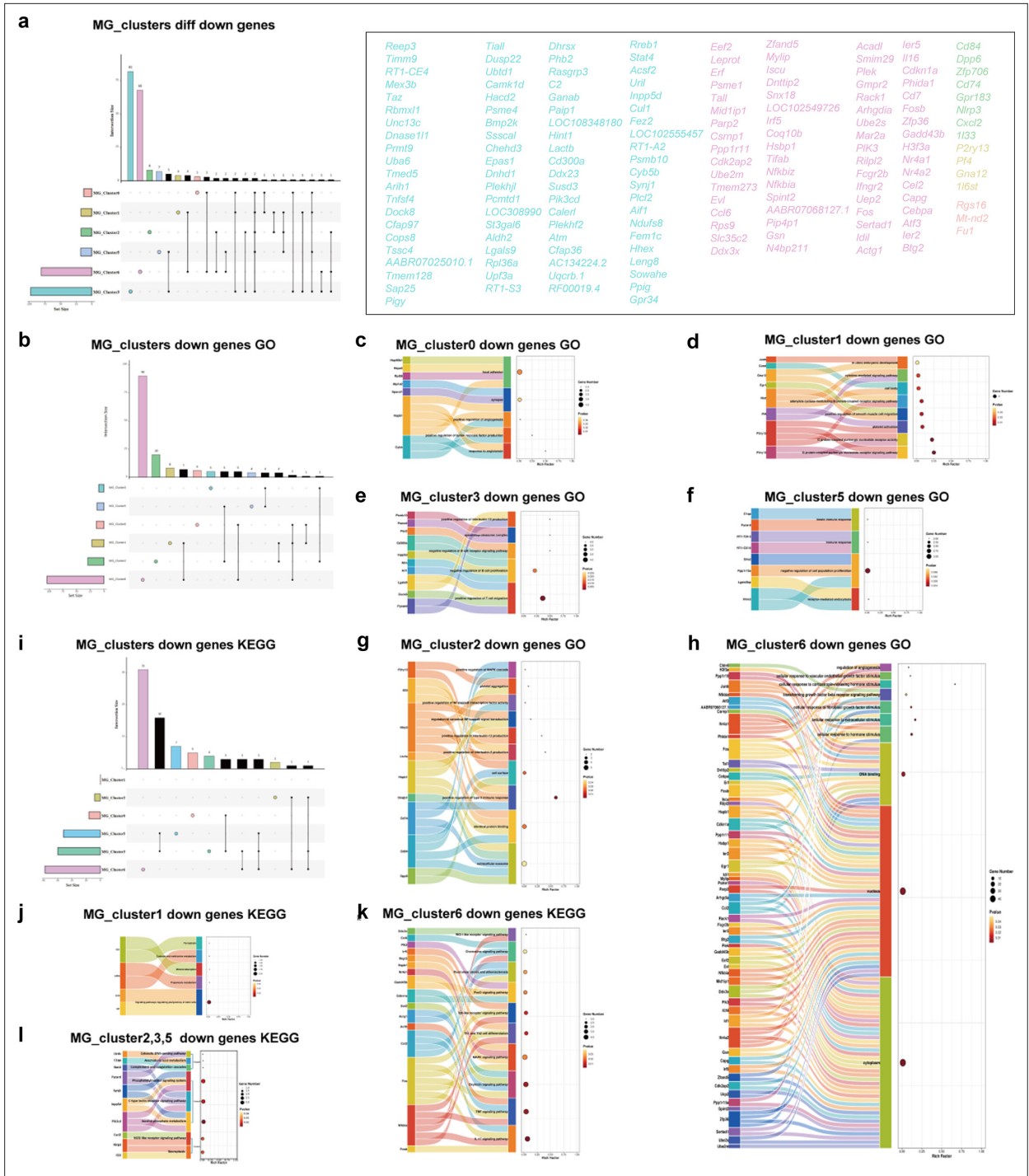

**Figure 15.** Functional enrichment analysis of unique downregulated genes in MG_clusters after electroacupuncture (EA). (**a**) Crossover and separation of downregulated genes in six clusters of microglia. The histograms with different colors represent different clusters and gene numbers. Colored circles represent genes that are only downregulated in the corresponding clusters, and these genes are listed in their respective colors. (**b, i**) Same and different GO and KEGG for down-regulated gene enrichment in six clusters of microglia. (**c–h**) Unique key GO entries and their genes in the six clusters are displayed. (**j–l**) Unique key KEGG pathways and their genes in the five clusters (MG_Cluster0 has no unique enrichment pathway). GO, Gene Ontology; KEGG, Kyoto Encyclopedia of Genes and Genomes.

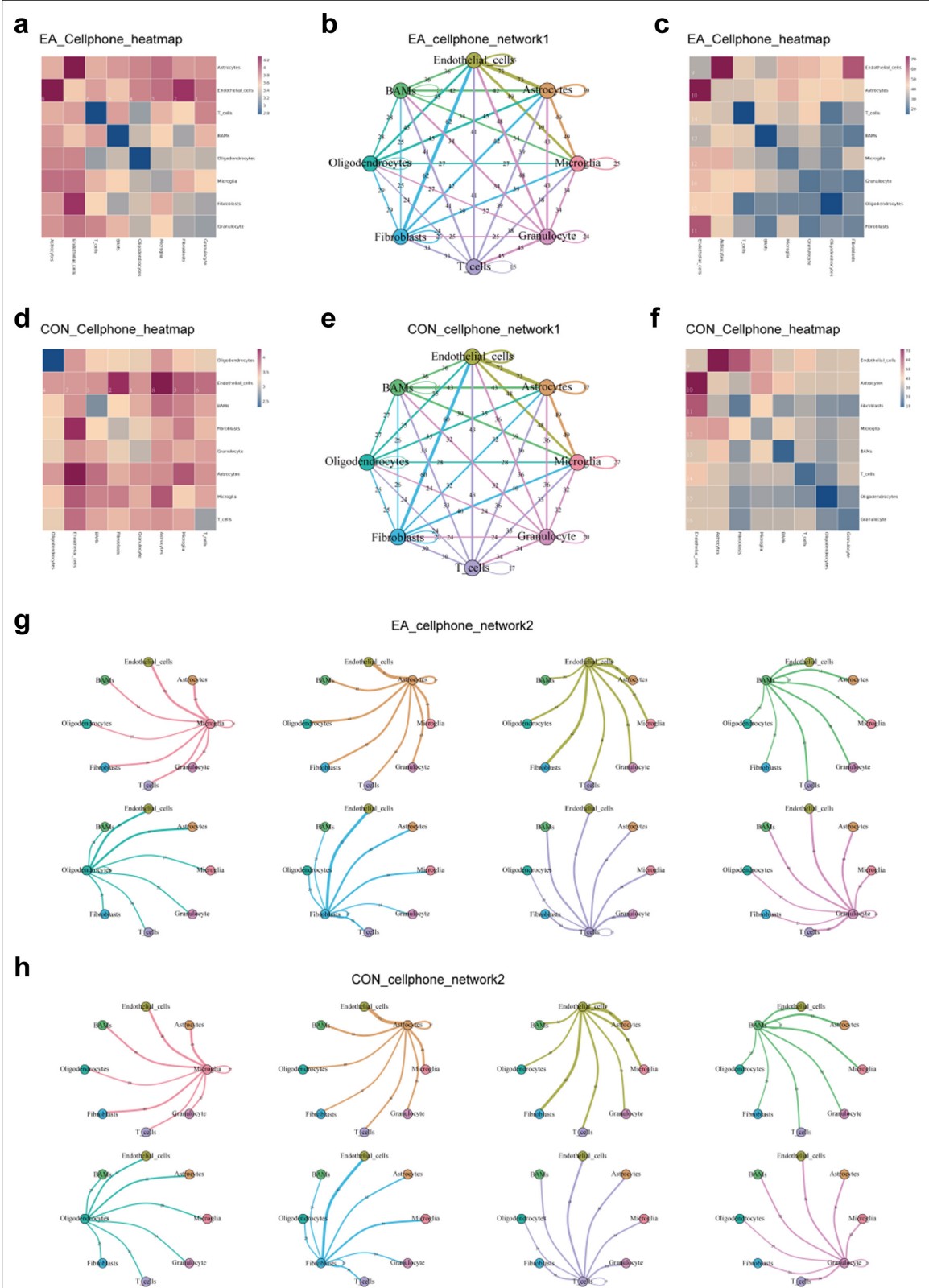

**Figure 16.** The interactions between cells, related to the section 'Coordination of cell communication in the blood-brain barrier (BBB) following EA intervention'. (**a**–**c**, **f**) Heatmap of the density of ligand-receptor pairs in the EA and CON groups. The redder the color, the closer the relationship between the cell groups; (**c**) and (**b**) are the original values; (**a**) and (**f**) are the values after taking the log value. (**b, e, g, h**) Network diagram of the active degree of the ligand-receptor relationship between two cells. (**b**) and (**e**) show the overall interaction network of the two groups. (**g**) and (**h**) show

*Figure 16 continued*

interactions of individual cells with other cells in both groups. The thicker the connection, the closer the correlation between the cells on the surface. CON, control; EA, electroacupuncture.

and SEMA4A_PLXND1 were only present in the CON group. The p-value of TGFB1_TGFBR3 was smaller in the EA group, while the *P*-value of CXCR6_CXCL16 was smaller in the CON group, and three pairs had a larger mean in the EA group (*Figure 17g*). Finally, EA activated the communication of JAG1_NOTCH4 and TNFSF12_TNFRSF25 and blocked the communication of IGFBP3_TMEM219 and MERTK_GAS6 in BAM-endothelial cells (*Figure 17h*).

## Discussion

Increased expression of inflammatory markers such as C-C motif chemokine ligand (CCL-4 and CCL-3) and TMEM119 and decreased expression of cellular communication network family proteins results in increased BBB permeability (*Lee et al., 2017*; *You et al., 2019*; *Davidson et al., 2022*; *Liu et al., 2022*; *Liu et al., 2024*). Previous studies have shown that the AAV-PHP.eB delivery is facilitated by the receptor protein lymphocyte antigen-6E (LY6E) in humans (*Fu et al., 2021*). EA pretreatment reduces BBB permeability and upregulates the expression of *Irf7* in healthy rats (*Wu et al., 2019*). Moreover, functional analysis in our study identified numerous terms and pathways associated with the BBB. Unexpectedly, the large clustering differences in microglia with and without EA intervention caught our attention. Indeed, microglia may serve as crucial priming cells for EA-induced opening of the BBB. Consequently, future research could focus on this aspect, and our laboratory is currently investigating this phenomenon, even though current research on microglia and BBB is mostly focused on the pathology and neuroinflammation of neurodegenerative diseases (*Xu et al., 2015*).

Higher basal Hmox1 expression in microglia is accompanied by increased oxidative stress, which directly affects the permeability of the BBB. Furthermore, Hmox1 regulates microglial polarization to the M2 phenotype (*Groh et al., 2023*). Of the top 10 genes that were downregulated after EA treatment, *Mx1*, *Mx2*, *Ifi27*, and *Ly6e* are immune-related genes, while Iba1 (encoded by *Aif1*) is a canonical microglial marker (*Gandal et al., 2018*). In addition, Pycard and Ltc4s function as mediators in apoptosis and inflammation (*McConnell and Vertino, 2000*; *Agostini et al., 2004*; *Faustin et al., 2007*; *Fernandes-Alnemri et al., 2007*; *Bryan et al., 2009*; *Hasegawa et al., 2009*; *Hornung et al., 2009*; *Taxman et al., 2011*; *Lu et al., 2014*; *Guan et al., 2015*; *Shen et al., 2019*; *Shen et al., 2021*).

The relationship between FK506 binding protein (FKBP) and the immunosuppressive drug FK506 has been examined; FKBP5 and FKBP4 are associated with neurological diseases, including Parkinson's disease and AD (*McConnell and Vertino, 2000*; *Bryan et al., 2009*; *Jiang et al., 2023*). Notably, the basic neural basis synapses of memory and cognitive function are highly susceptible to synaptic phagocytosis by activated microglia and may improve cognitive function by reducing excessive activation of microglia (*Reverte et al., 2024*). We also observed enrichment of the pathway associated with cocaine addiction, which has been associated with microglia for synaptic adaptations in the nucleus accumbens synapses during cocaine withdrawal (*Victor et al., 2022*). Another pathway, thyroid hormone synthesis, is important for the microglial phenotype (*Figures 12d, 13b and c*, *Supplementary file 23* and *Supplementary file 24*). The trigger receptor expressed in myeloid cell 2 (TREM2) is a cell surface receptor found in macrophages and microglia that responds to disease-related signals to regulate the phenotype of these innate immune cells. TREM2 is regulated by thyroid hormones, and its expression in macrophages and microglia is stimulated by thyroid hormones and synthetic thyroid hormone agonists (thyroid inhibitors).

In addition to considering the overall clustering of endothelial cells and microglia, which are cells with large differences in clustering, subsequent experiments should also focus on changes in the characteristics of the clustering of these cell subpopulations to explore whether there are certain unknown potential key subpopulations that play a critical role in BBB permeability. Similarly, cell communication analysis should not be confined to astrocytes and pericytes, which are typically regarded as cell types that are functionally associated with the BBB or comprised in the BBB. Differences in clustering and gene expression were also present in fibroblasts, BAMs, and T cells, although not so pronounced as to be focused on in this article, but it is possible that they are also involved in the regulation of BBB permeability.

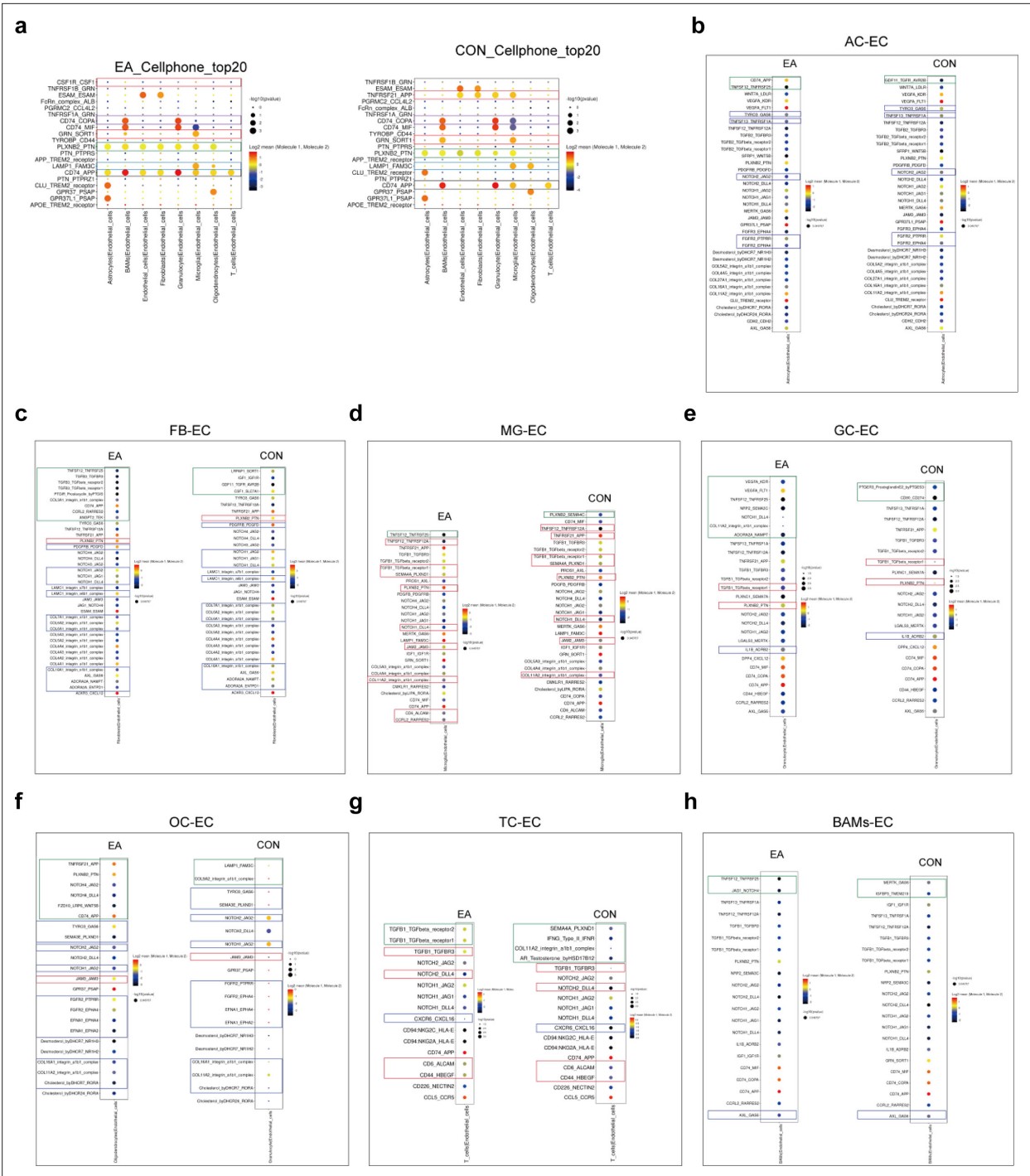

**Figure 17.** Cell communication with endothelial cells, related to the section 'Coordination of cell communication in the blood-brain barrier (BBB) following electroacupuncture (EA) intervention'. (**a**) The top 20 ligand receptor pairs with mean value in the EA and CON groups, respectively. Different colored boxes are used to indicate pairs with differences between the groups. (**b–h**) Detailed ligand-receptor pairs for communication between other cells to endothelial cells. The depth of the colors in the image represents the strength of the corresponding interaction, and the size of the dots represents the significance of the p-value. The green box represents pairs that are only present in a specific group (EA or CON), the red box represents pairs that are upregulated in the EA group (mean or p-value), and the blue box represents pairs that are downregulated in the EA group. CON, control; EA, electroacupuncture.

This study had some limitations. First, as this investigation represents the initial analysis of single-cell transcriptome sequencing in the frontal cortex following EA intervention, there exists a limited understanding of the signaling pathways involved in EA-mediated BBB function. Second, a rat model was selected for this study due to considerations of EA operational precision. Third, additional protein analyses and functional validation are necessary to confirm the putative roles of each cellular phenotype. Finally, the possibility cannot be excluded that certain cells or cell types may be lost during tissue dissociation and cell enrichment; indeed, pericytes and neurons, which also play significant roles in the brain, were not enriched in this study. Nevertheless, this single-cell transcriptome sequencing in a rat model to investigate the target of action of EA-opening BBB could facilitate the study of BBB-opening-related mechanisms, which is crucial to numerous current investigations of exogenous drugs crossing the BBB into the brain.

## Conclusions

The study of the molecular and cellular mechanisms that govern the function of the BBB is complex and requires a thorough understanding of the classification and molecular characteristics of cells, as well as detailed characterization of the spatial organization and interaction of molecularly defined cell types. This is because the spatial relationship between cells is the primary determinant of intercellular interactions and communication, which are mediated by juxtacrine and paracrine signaling. Our study revealed a unique frontal cortex-specific transcriptome signature and elucidated the cell types and clusters that mediate EA-induced transcriptional changes, including cellular communication. Our study highlights the characteristic changes in brain endothelial cells under EA intervention, as well as the cell types that synergistically participate in the structural and functional changes of the BBB, such as microglia and astrocytes, providing a framework for the mechanism of EA opening the BBB. The development of high-resolution, spatially resolved whole-brain cell atlases will provide invaluable resources for investigating BBB function during EA interventions. Future research in this area is necessary to elucidate the intricate details of intercellular communication and gain a deeper understanding of the mechanisms underlying BBB function.

## Acknowledgements

We appreciate the support of the Key Laboratory of Acupuncture and Neurology of Zhejiang Province and the technical support from the Medical Research Center, Academy of Chinese Medical Sciences, Zhejiang Chinese Medical University. We thank Mr. Xushen Chen of the School of Public Health, Zhejiang Chinese Medical University, for his help with the language. We would also like to thank Editage (https://www.editage.cn/) for English language editing. This work was supported by the National Natural Science Foundation of China [Xianming Lin: grant numbers 82174502, 82474626]; Zhejiang provincial natural science foundation [Congcong Ma: grant number LQN25H270008]; Special Project of Modernization of Traditional Chinese Medicine in Zhejiang Province in 2022 [Xianming Lin: grant number 2022ZX009]; Scientific Research Project of the Affiliated Hospital of Zhejiang Chinese Medical University in 2022 [Xianming Lin: grant number 2022FSYYZZ10]; Scientific Research Project of the Affiliated Hospital of Zhejiang Chinese Medical University in 2023 [Congcong Ma: grant number 2023FSYYZQ11]. The funders had no role in the study design, data collection and interpretation, or the decision to submit the work for publication.

## Additional information

### Funding

| Funder | Grant reference number | Author |
|---|---|---|
| National Natural Science Foundation of China | 82174502 | Xianming Lin |
| National Natural Science Foundation of China | 82474626 | Xianming Lin |

| Funder | Grant reference number | Author |
|---|---|---|
| Zhejiang Provincial Natural Science Foundation | LQN25H270008 | Congcong Ma |
| Special Project of Modernization of Traditional Chinese Medicine in Zhejiang Province | 2022ZX009 | Xianming Lin |
| Scientific Research Project of the Affiliated Hospital of Zhejiang Chinese Medical University | 2022FSYYZZ10 | Xianming Lin |
| Scientific Research Project of the Affiliated Hospital of Zhejiang Chinese Medical University | 2023FSYYZQ11 | Congcong Ma |

The funders had no role in study design, data collection and interpretation, or the decision to submit the work for publication.

### Author contributions

Congcong Ma, Conceptualization, Data curation, Investigation, Visualization, Writing – original draft, Writing – review and editing; Zhaoxing Jia, Data curation, Investigation, Methodology; Tianxiang Jiang, Investigation, Methodology, Project administration; Qian Cai, Software, Investigation, Methodology, Project administration; Jinding Yang, Data curation, Formal analysis, Project administration; Lin Gan, Software, Validation; Kecheng Qian, Methodology; Zixin Pan, Qinyu Ye, Investigation; Mengyuan Dai, Supervision, Investigation, Methodology, Project administration; Xianming Lin, Supervision, Funding acquisition, Project administration, Writing – review and editing

### Author ORCIDs

Xianming Lin ⓘ https://orcid.org/0000-0002-2729-8324

### Ethics

The Animal Ethics Committee of Zhejiang Chinese Medical University approved the experimental procedures (Approval No.: IACUC-20220507-01). All experiments were performed in accordance with the ARRIVE guidelines. Consent to participate was not applicable.

Joint Public Review: https://doi.org/10.7554/eLife.107938.3.sa1
Author response https://doi.org/10.7554/eLife.107938.3.sa2

## Additional files

### Supplementary files

Supplementary file 1. Pathway analysis for genes downregulated by SMES only in EC_cluster0.

Supplementary file 2. Pathway analysis for genes upregulated by SMES only in EC_cluster0.

Supplementary file 3. Pathway analysis for genes upregulated only in EC_cluster2.

Supplementary file 4. Pathway analysis for genes downregulated only in EC_cluster2.

Supplementary file 5. Pathway analysis for genes downregulated only in EC_cluster4.

Supplementary file 6. Pathway analysis for genes upregulated only in EC_cluster4.

Supplementary file 7. Pathway analysis for genes downregulated only in EC_cluster5.

Supplementary file 8. Pathway analysis for genes upregulated only in EC_cluster5.

Supplementary file 9. Gene Ontology (GO) analysis for MG_cluster1 top genes only ($S \geq 2$).

Supplementary file 10. Kyoto Encyclopedia of Genes and Genomes (KEGG) analysis for MG_cluster1 top genes only (counts top 20).

Supplementary file 11. Kyoto Encyclopedia of Genes and Genomes (KEGG) analysis for MG_cluster1 top genes only (20 smallest p-values).

Supplementary file 12. Gene Ontology (GO) analysis for MG_cluster0 top genes only (S≥2).

Supplementary file 13. Kyoto Encyclopedia of Genes and Genomes (KEGG) analysis for MG_cluster0 top genes only (counts top 20).

Supplementary file 14. Kyoto Encyclopedia of Genes and Genomes (KEGG) analysis for MG_cluster0 top genes only (20 smallest p-values).

Supplementary file 15. Gene Ontology (GO) analysis for MG_cluster2 top genes only (S≥2).

Supplementary file 16. Kyoto Encyclopedia of Genes and Genomes (KEGG) analysis for MG_cluster2 top genes only (counts top 20).

Supplementary file 17. Kyoto Encyclopedia of Genes and Genomes (KEGG) analysis for MG_cluster2 top genes only (20 smallest p-values).

Supplementary file 18. Gene Ontology (GO) analysis for MG_cluster3 top genes only (S≥2).

Supplementary file 19. Gene Ontology (GO) analysis for MG_cluster5 top genes only (S≥2).

Supplementary file 20. Kyoto Encyclopedia of Genes and Genomes (KEGG) analysis for MG_cluster5 top genes only (counts top 20).

Supplementary file 21. Kyoto Encyclopedia of Genes and Genomes (KEGG) analysis for MG_cluster5 top genes only (20 smallest p-values).

Supplementary file 22. Gene Ontology (GO) analysis for MG_cluster6 top genes only (S≥2).

Supplementary file 23. Kyoto Encyclopedia of Genes and Genomes (KEGG) analysis for MG_cluster3 top genes only (counts top 20).

Supplementary file 24. Kyoto Encyclopedia of Genes and Genomes (KEGG) analysis for MG_cluster3 top genes only (20 smallest p-values).

MDAR checklist

## Data availability

This study did not generate new unique reagents. The scRNA-seq data files are publicly available in the Gene Expression Omnibus under accession number GSE272895.

The following dataset was generated:

| Author(s) | Year | Dataset title | Dataset URL | Database and Identifier |
|---|---|---|---|---|
| Ma C, Jia Z, Gan L, Lin X | 2024 | Profiling the rat cortical single-cell transcriptome reveals the potential mechanism of specific mode electroacupuncture stimulation opening the blood-brain barrier | https://www.ncbi.nlm.nih.gov/geo/query/acc.cgi?acc=GSE272895 | NCBI Gene Expression Omnibus, GSE272895 |

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
