## [Editor Report · eLife Assessment]

This study presents a **valuable** finding that the blood-brain barrier (BBB) may be modulated through specific modes of electroacupuncture stimulation. The data were collected and analyzed using a **solid** and validated methodology, and can be used as a starting point for functional studies of the BBB for drug delivery across healthy and diseased states. The work will be of broad interest to scientists working in the field of drug delivery and drug development.

---

## [Referee Report · Joint Public Review]

This study employs single-cell RNA sequencing to investigate how electroacupuncture (EA) stimulation alters the transcriptional profiles of central nervous system cell types following blood-brain barrier (BBB) opening. The authors seek to characterize changes in gene expression and pathway activities across diverse neural cells in response to electroacupuncture (EA) stimulation using high-resolution transcriptomics. This approach has the potential to elucidate the cellular mechanisms underlying EA stimulation and their implications for therapeutic intervention. The work engages with a timely and biologically significant question regarding noninvasive stimulation methods to manipulate BBB permeability. However, no in vivo/in vitro functional assays are provided to validate the changes in BBB permeability or cytokine release in the tested models. The experimental rationale remains inadequately explained, and key details regarding the magnitude, duration, and spatial distribution of BBB opening in this system are still lacking.

---

## [Author Response]

The following is the authors’ response to the original reviews.

**Reviewer #1 (Public review):**
Summary:The work from this paper successfully mapped transcriptional landscape and identified EA-responsive cell types (endothelial, microglia). Data suggest EA modulates BBB via immune pathways and cell communication. However, claims of "BBB opening" are not directly proven (no permeability data).(1) No in vivo/in vitro assays confirm BBB permeability changes (e.g., Evans blue leakage, TEER).(2) Only male rats were used, ignoring sex-specific BBB differences.(3) Pericytes and neurons, critical for the BBB, were not captured, likely due to dissociation artifacts.(4) Protein-level validation (Western blot, IHC) absent for key genes (e.g., LY6E, HSP90).(5) Fixed stimulation protocol (2/100 Hz, 40 min); no dose-response or temporal analysis.

We sincerely apologize for the oversight regarding the description of changes in blood-brain barrier permeability. In fact, our team conducted a series of preliminary studies that verified this aspect, and we hace provided a more detailed introduction in the introduction section, in lines 60-71 of the manuscript.

We are very grateful to the reviewers for pointing out the important and meaningful issue of "gender-specific BBB differences." We will make this a focal point in our future research.

As for pericytes and neurons, we acknowledge their importance in the function of the blood-brain barrier. We acknowledge the importance of pericytes and neurons in the blood-brain barrier. However, neurons are absent because our sample processing method involves dissociation. During the dissociation procedure, neuronal axons, which are relatively long, are filtered out during the frequent cell suspension steps and cannot enter the downstream microfluidic system for analysis, so they are not present in our data. Since this experiment is primarily focused on non-neuronal cells, we did not choose to use nucleus extraction for sample processing. As for pericytes, we believe they are not captured because their proportion in our samples is extremely low, which is why they are not present in the data. Further research may require single-nucleus transcriptomics or the separate isolation of these two cell types for study. Of course, in our current mechanistic studies, we are also fully considering the important roles these two cell types play in BBB function.

In addition, to validate the results at the protein level, we have recently conducted some experiments. However, as several proteins are currently at a critical stage of further experimental validation, it is not appropriate to present them in the manuscript at this time. Instead, we have uploaded the relevant data as an appendix for your review. This includes a figure of several protein markers we examined, as well as a table of the antibodies used.

This section is also further elaborated in the introduction and its references.

**Reviewer #2 (Public review):**
Summary:This study uses single-cell RNA sequencing to explore how electroacupuncture (EA) stimulation alters the brain's cellular and molecular landscape after blood-brain barrier (BBB) opening. The authors aim to identify changes in gene expression and signaling pathways across brain cell types in response to EA stimulation using single-cell RNA sequencing. This direction holds promise for understanding the consequences of noninvasive methods of BBB opening for therapeutic drug delivery across the BBB.(1) The work falls short in its current form. The experimental design lacks a clear justification, and readers are not provided with sufficient background information on the extent, timing, or regional specificity of BBB opening in this EA model. These details, established in prior work, are critical to understanding the rationale behind the current transcriptomic analyses.(2) Further, the results are often presented with minimal context or interpretation. There is no model of intercellular or molecular coordination to explain the BBB-opening process, despite the stated goal of identifying such mechanisms. The statement that EA induces a "unique frontal cortex-specific transcriptome signature" is not supported, as no data from other brain regions are presented. Biological interpretation is at times unclear or inaccurate - for instance, attributing astrocyte migration effects to endothelial cell clusters or suggesting microglial tight junction changes without connecting them meaningfully to endothelial function.(3) The study does include analyses of receptor-ligand signaling and cell-cell communication, which could be among its most biologically rich outputs. However, these are relegated to supplementary material and not shown in the leading figures. This choice limits the utility of the manuscript as a hypothesis-generating resource.(4) Overall, while the dataset may be of interest to BBB researchers and those developing technologies for drug delivery across the BBB, the manuscript in its current form does not yet fulfill its interpretive goals. A more integrated and biologically grounded analysis would be beneficial.

This section is also further elaborated in the introduction and its references.

Our current study is actually based on previous findings that electroacupuncture can open the BBB, with a more pronounced effect observed in the frontal lobe (this aspect should be further described in the research background). Building on this foundation, our aim is to delineate the potential biological mechanisms involved. Therefore, we selected frontal lobe tissue as our primary choice for sequencing and have not yet investigated differences across other brain regions, although this may become a focus of future research. Additionally, we recognize that the mechanism underlying BBB opening is complex, and at present, we cannot determine whether it is driven by a single direct factor or by coordinated actions between cells or molecules. As such, our results are presented only briefly for now, and we will carefully consider whether to supplement our findings by incorporating insights from other studies.

Considering the overall data layout and the length of the article, we ultimately decided not to make any changes to the presentation of the article's data. The images included in the supplementary materials are also thoroughly described and referenced in the manuscript, allowing readers to selectively view any data they are interested in.

Indeed, our current dataset and analysis tend to present objective data results. We are also conducting a series of validations that may be related to the biology of the blood-brain barrier, and we look forward to sharing and discussing any future research findings with you and everyone.

**Reviewer #1 (Recommendations for the authors):**
(1) Figures 3-7: Label treatment groups (CON vs. EA) consistently in legends.(2) Methods: Specify rat strain (Sprague-Dawley) in the abstract.(3) Clarify Limitations: Explicitly state that BBB opening is inferred, not proven.

This section has been revised at lines 743-733, 748, 949, 754-755, and 759-760 of the manuscript.

Revised at line 31 of the manuscript.

Thank you for your feedback. The background information on the open evidence of BBB has been added to the introduction.

**Reviewer #2 (Recommendations for the authors):**
(1) Abstract and Introduction• Include specific key findings in the abstract to improve clarity and reader engagement.• Expand the introduction to situate this work in the context of other BBB-opening methods (e.g., ultrasound) and the known consequences of BBB disruption.• Clarify the rationale for choosing electroacupuncture.• Include information (perhaps summarized from previous studies) about the extent, timeline, and functional assessment of BBB opening in this model to help justify the single-cell RNA-seq design.(2) Experimental Rationale and Context• Reiterate experimental design and rationale in each results section, rather than relying exclusively on the Methods section.• Specify the time point of tissue collection relative to the EA intervention.• Describe the anatomical sites of acupuncture stimulation and their physiological relevance.(3) Data Presentation• Replace the human brain cartoon in Figure 1 with an anatomically appropriate rat brain schematic.• Reevaluate which data are presented in the main versus supplementary figures. Highlight biologically meaningful results, such as cell-cell communication and ligand-receptor interactions, in the main figures rather than supplementary data.(4) Interpretation and Modeling• More carefully link transcriptional changes (e.g., Wnt signaling in microglia) to biologically plausible mechanisms of BBB regulation-e.g., microglial signaling to endothelial cells.• Clarify whether the presence of granulocytes and T cells might result from a lack of perfusion prior to brain dissection.• Consider proposing a model (even speculative) of how EA leads to BBB opening based on observed transcriptional changes.

First, for the sake of brevity in the abstract, we did not present specific results in this section. Second, since BBB opening via EA is a unique strategy, our previous studies have examined the opening time window and the recovery of the BBB after EA intervention (as mentioned in the introduction). We believe its characteristics differ from those of ultrasound-induced BBB opening and BBB disruption, so we did not conduct comparative discussions, but objectively presented our research findings. In further functional validation experiments, we may consider integrating other opening strategies in our studies. Additionally, the choice of electroacupuncture was based on our previous series of studies, which have already been outlined in the research background. Finally, we did indeed determine the experimental design of this study based on prior research, as described in the background section of the introduction.

We decided not to make changes to this section in the manuscript after careful consideration. The setup of electroacupuncture intervention and controls has been thoroughly discussed in our previous studies (as referenced in the introduction), so we have not repeated it in this manuscript. Overall, building on all our previous findings, this study focuses primarily on the potential mechanisms of EA intervention. The anatomical sites of acupuncture stimulation and their physiological relevance are another key area of our research, and we are currently conducting a series of related studies. We look forward to sharing these findings with you in the future.

We have already changed the human brain diagram in Figure 1 to a rat brain diagram, and have replaced Figure 1 in the files with the revised version. However, considering the overall data layout and the length of the article, we ultimately decided not to make changes to the data presentation in the manuscript. The images in the supplementary materials are also thoroughly described and referenced in the manuscript, allowing readers to selectively view the data they are interested in.

This section has provided us with excellent suggestions for further exploration, although no changes have been made to the manuscript at this time. In the future, we may conduct more detailed transcriptomic studies focusing on sex differences and different brain regions, which will allow for a more comprehensive analysis of the biological mechanisms involved in BBB regulation.